# Contributions of Nordic anthropogenic emissions on air pollution and premature mortality over the Nordic region and the Arctic

Ulas Im[1,2], Jesper H. Christensen[1,2], Ole-Kenneth Nielsen[1,2], Maria Sand[3], Risto Makkonen[4,5], Camilla Geels[1,2], Camilla Anderson[6], Jaakko Kukkonen[4], Susana Lopez-Aparicio[7], Jørgen Brandt[1,2]

Aarhus University, Department of Environmental Science, Atmospheric Modelling Section, Frederiksborgvej 399, Roskilde, Denmark.
Interdisciplinary Center for Climate Change (iCLIMATE), Frederiksborgvej 399, Roskilde, Denmark.
Center for International Climate Research, Postboks 1129 Blindern, 0318 Oslo, Norway.
Finnish Meteorological Institute, Erik Palmenin aukio 1, P.O.Box 503, FI-00101, Helsinki, Finland.
University of Helsinki, Institute for Atmospheric and Earth System Research, P.O. Box 64, 00014, Helsinki, Finland.
Swedish Meteorological and Hydrological Institute, SE-60176 Norrköping, Sweden.
NILU - Norwegian Institute for Air Research, Instituttveien 18, P.O. Box 100, 2027 Kjeller, Norway.

Abstract

This modelling study presents the sectoral contributions of anthropogenic emissions in the four Nordic countries; Denmark, Finland, Norway and Sweden, on air pollution levels and the associated health impacts and costs over the Nordic and the Arctic region for the year 2015. The Danish Eulerian Hemispheric Model (DEHM) has been used on a 50 km resolution over Europe in tagged mode in order to calculate the response of a 30% reduction of each emission sector in each Nordic country individually. The emission sectors considered in the study were energy production, non-industrial/commercial heating, industry, traffic, off-road mobile sources, and waste management/agriculture. In total, 28 simulations were carried out. Following the air pollution modelling, the Economic Valuation of Air Pollution (EVA) model has been used to calculate the associated premature mortality and their costs. Results showed that more than 80% of the $PM_{2.5}$ concentration was attributed to transport from outside these four countries, implying an effort outside the Nordic region in order to decrease the pollutant levels over the area. The leading emission sector in each country was found to be non-industrial combustion (contributing by more than 60% to the total $PM_{2.5}$ mass coming from the country itself), except for Sweden, where industry contributed to $PM_{2.5}$ with a comparable amount as non-industrial combustion. In addition to non-industrial combustion, the next most important source categories were industry, agriculture and traffic. The main chemical constituent of $PM_{2.5}$ concentrations that comes from the country itself is calculated to be organic carbon in all countries, which suggested that non-industrial wood burning was the dominant national source of pollution in the Nordic countries. We have estimated the total number of premature mortality cases due to air pollution to be around 4 000 in Denmark and Sweden and around 2 000 in Finland and Norway. These premature mortality cases led to a total cost of 7 billion Euros in the selected Nordic countries. The assessment of the related premature mortality and associated cost estimates suggested that non-industrial combustion, together with industry and traffic, will be the main sectors to be targeted in emission mitigation strategies in the future.

Introduction

Air pollution is the world's largest single environmental health risk (WHO, 2014), estimated to be responsible for 3.7 million premature deaths in 2012 from urban and rural sources worldwide. In Europe, recent results (Andersson et al., 2009; Brandt et al., 2013a; 2013b; Geels et al., 2015; Im et al., 2018a; Liang et al., 2018; Solazzo et al., 2018) show that outdoor air pollution causes ~500 000 premature deaths in Europe. Brandt et al. (2013a) calculated that due to exposure to ambient air pollution, there were around 3.500 premature deaths in 2011 in Denmark alone. Lehtomäki et al. (2018) have recently evaluated that ambient air pollution caused approximately 2000 premature deaths in Finland in 2015. Other studies have made assessments for some of the Nordic countries (Denmark, Sweden and Finland) with estimates ranging from 6500 to 9500 for the year 2000 (Geels et al., 2014; Watkiss et al., 2005, Karvosenoja et al., 2010, respectively). Kukkonen et al. (2018) and Forsberg et al. (2015) have concluded that long-range transported fine particulate matter dominates the health effects in the Nordic countries, with largest contribution to long-term effects in Sweden originating from south-western Europe, while the largest contribution to short-term exposure originates from south-eastern Europe (Jönsson et al. 2013).

Air pollution is a transboundary problem covering global, regional, national and local sources, leading to large spatial variability and therefore to large differences in the geographical distribution of human exposure to air pollution (Im et al., 2018a,b). In the Nordic countries, there are large spatial differences in air pollution levels because of long-range transported and polluted air masses especially from the south and east as well as due to the degree of urbanization. There are also local differences depending on wind direction and distance from local emission sources such as road transport, power plants and industry (Brandt et al., 2013a). Furthermore, the widespread use of domestic wood stoves in the Nordic countries represents a special challenge for exposure to air pollution (Kukkonen et al., 2019) , where e.g. more than a third of the health impacts from Danish emissions are due to smoke from wood stoves. International ship traffic is also a significant source of air pollution and health impacts in highly trafficked areas of the Baltic and North Seas (Brandt et al., 2013b; Jalkanen et al., 2016, Johansson et al., 2017). Based on simulations for the period 1997-2003, Andersson et al. (2009) calculated that Sweden contributed to 1.4% of the European Primary $PM_{2.5}$ ($PPM_{2.5}$) mass concentrations while Denmark, Finland and Norway were responsible for 4% of European $PPM_{2.5}$. Contribution to secondary inorganic aerosol (SIA) levels were much smaller (0.5% from Sweden and 1.4% from Denmark, Finland and Norway). They also calculated a death rate increase of 2 and 3% due to exposure to $PPM_{2.5}$ and SIA, respectively, in Europe due to emissions from Denmark, Finland, Norway and Sweden.

The external (or indirect) costs to society related to health impacts from air pollution are substantial. In the whole of Europe, the total external costs have been estimated to be approx. 800 billion Euros per year and in Denmark alone the external costs are nearly 4 billion Euro per year (Brandt et al., 2013a). In a more recent study, Im et al. (2018a), using a multi-model ensemble of 14 chemistry transport models (CTM), estimated that ambient air pollution in Europe in 2010 was responsible for 414 000±100 000 premature deaths, leading to a cost of 300 billion Euros. The study also showed that a 20% decrease of anthropogenic emissions in Europe source could avoid 47 000 premature deaths in Europe, while a similar reduction in the U.S. would avoid around 1 000 premature deaths in Europe due to long-range transport.

The Nordic countries are generally characterized among the EU countries with low air pollution levels (EEA, 2018). $PM_{2.5}$ levels are below the EU legislated limit value of 25 µg m$^{-3}$ as well as the

WHO limit value of 10 μg m$^{-3}$ (EEA, 2018). However, there are still large impacts of air pollution
on human health and climate in the region itself (Arctic Council, 2011; Brandt et al., 2013a;
Forsberg et al., 2015), as well as over the Arctic (Sand et al., 2015). The Task Force on Short Lived
Climate Forcers of the Arctic Council reported that measures aimed at decreasing Nordic emissions
will have positive health effects for communities exposed to air pollution. In a recent study, Sand et
al. (2015) showed that although the largest Arctic warming source is from Asian emissions, the
Arctic is most sensitive, per unit mass emitted, to Short Lived Climate Forcers (SLCF) emissions
from a small number of activities within the Arctic nations themselves.
The aim of the study is to quantify the contributions of the main emission sectors in each of the
Nordic countries to air pollutant levels and their impacts on premature mortality and associated
costs in the Nordic region and the Arctic. This will help us identify the emission sectors in these
Nordic countries that should be targeted for mitigation to decease the air pollution and exposure
levels in the Nordic countries, that are originated within the region. In addition, we also aim to give
a first estimate of the impact of transported air pollution on the Arctic population. In order to
achieve this, we have coupled the Danish Eulerian Hemispheric Model (DEHM) to the Economic
Valuation of Air Pollution (EVA) model and conducted a number of perturbation simulations
targeting different emission sectors in the four Nordic countries; Denmark, Finland, Norway and
Sweden, for the year 2015. Year 2015 is selected to be in agreement with the ongoing Coupled
Model Intercomparison Project Phase 6 (CMIP6: Eyring et al., 2016), where the current year is
2015. As the present study will also look at the impacts in the future using baseline scenarios from
the CMIP6, we have selected the present year to be 2015 for consistency. The models and
perturbation simulations are described in Section 2, the model evaluation against surface
measurements in the Nordic countries are presented in Section 3.1, the contributions of sectoral
emissions on the air pollution levels in the Nordic region and the Arctic are presented in Section
3.2., and the health impacts and associated costs are presented in Section 3.3. Conclusions are given
in Section 4.
1.  Materials and methods
2.1. Danish Eulerian Hemispheric Model (DEHM)
The DEHM model was originally developed mainly to study the transport of SO2 and SO4 to the
Arctic (Christensen 1997), but has been extended to different applications during the last decades. It
has been documented extensively in Brandt et al. (2012) and evaluated in several intercomparison
studies (e.g. Solazzo et al., 2012 a,b; Solazzo et al., 2017; Im et al., 2018a,b) and recently joined the
suit of operational models in the Copernicus Atmospheric Monitoring System (CAMS) to provide
regional forecasts of air pollution over Europe. The DEHM model uses a 150 km×150 km spatial
resolution over the Northern Hemisphere, then nests to 50 km×50 km resolution over Europe,
extending up to 100 hPa through 29 vertical levels, with the first layer height of approximately 20
m. The meteorological fields were simulated by the Weather Research and Forecast Model (WRF,
Skamarock et al., 2008) setup with identical domains and resolution. The time resolution of the
DEHM model is one hour. The gas-phase chemistry module includes 58 chemical species, 9
primary particles, including natural particles such as sea-salt and 122 chemical reactions (Brandt et
al., 2012). The model also describes atmospheric transport and chemistry of lead, mercury, $CO_2$, as
well as POPs. Secondary organic aerosols (SOA) are calculated using the Volatility Base System
(VBS: Bergstrom et al., 2012). In addition to the anthropogenic PM and SOA due to biogenic
emissions, DEHM model also calculates sea-salt emissions and their transport and interactions with
other pollutants. The current version of the DEHM model does not include wind-blown or re-
suspended dust emissions. DEHM model does not output a $PM_{2.5}$ or $PM_{10}$ diagnostic, however
these are calculated off-line, using all anthropogenic and natural components of PM, in order to be
used in the health impact assessment described in Section 2.2.

In the current study, the DEHM model used anthropogenic emissions from the EDGAR-HTAP
database and biogenic emissions are calculated online based on the MEGAN model. The total
emission per country for the different pollutants are presented in Table 1. The sectoral distributions
of emissions in each country are presented in Figure 1. As seen in the Table 2, most SNAP
(Selected Nomenclature for Air Pollutants; CEIP, 2019) sectors are considered individually, while
some are merged in order to reduce the computational costs. All sectors in relation to industrial
activities (combustion, processes, solvent use and extraction and transport of fossil fuels) are
merged into an "Industry" source sector, while waste management and agriculture sectors were
lumped into one source sector.

As seen in Figure 1, non-industrial combustion (orange bars), where non-industrial combustion
dominates, stands out as a major source contributing to CO and PM emissions while industry (grey
bars) (Table 2) is the largest source of NMVOCs, NOx and SOx. Traffic (yellow bars) also
contributes significantly to CO and NOx. The largest source of $NH_3$ is from agriculture in
particular, as well as waste management (green bars) (Table 2).

Table 1. Total pollutant emissions in the Nordic countries (in Gg) in 2015.

|     | CO  | $NH_3$ | NMVOC | NOx | $SO_x$ | $PM_{10}$ | $PM_{2.5}$ |
|-----|-----|--------|-------|-----|--------|-----------|------------|
| DK  | 251 | 75     | 106   | 102 | 9      | 31        | 20         |
| FI  | 302 | 31     | 85    | 128 | 41     | 31        | 19         |
| NO  | 378 | 28     | 155   | 133 | 16     | 35        | 27         |
| SE  | 413 | 54     | 159   | 129 | 18     | 37        | 18         |



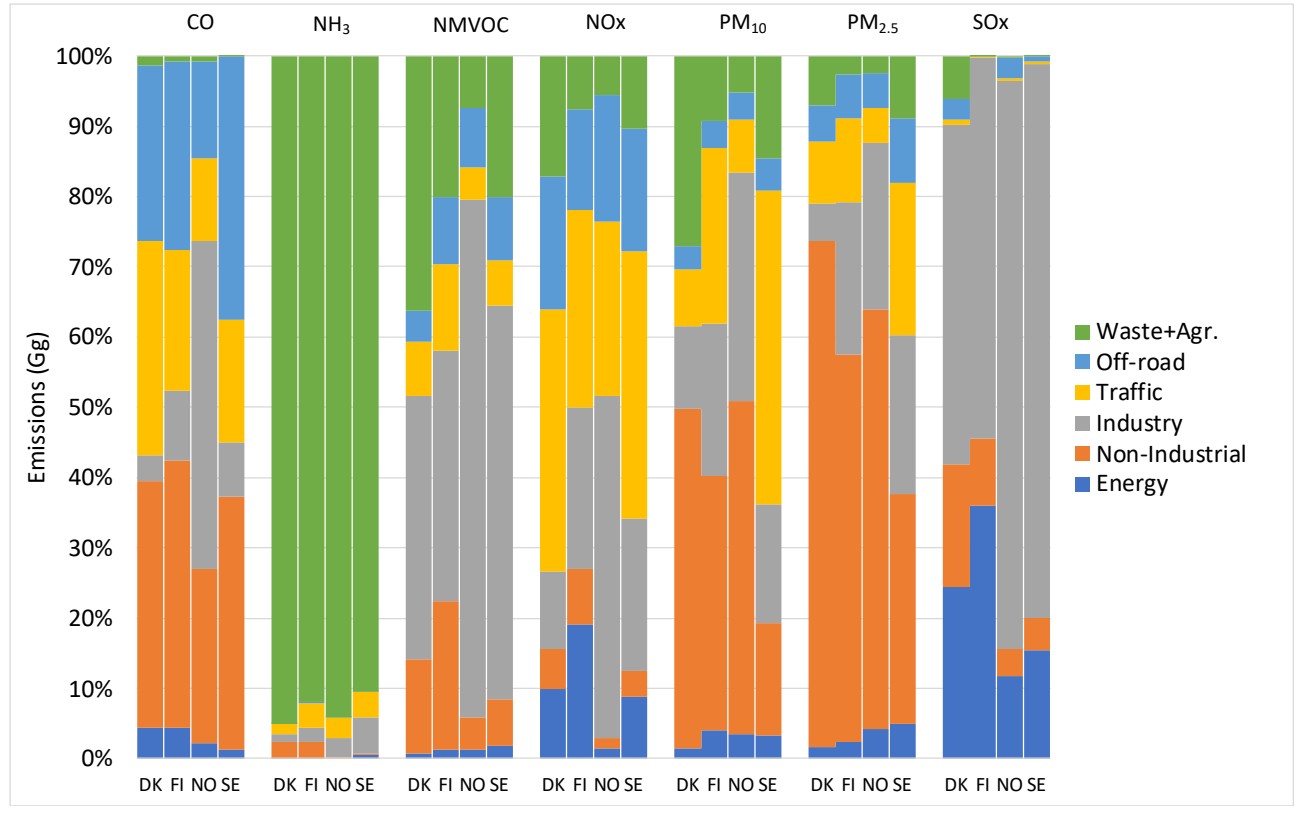

Figure 1. Relative distributions (%) of sectoral emissions of major air pollutants in the Nordic
countries.
2.1.1. Tagging Method
The tagging method keeps track of contributions to the concentration field from a particular
emission source or sector, as explained in detail in Brandt et al. (2013a). Tagging involves
modelling the background concentrations and the δ-concentrations (the contributions from a
specific emission source or sector to the overall air pollution levels) in parallel (as two different
runs under the same run), where special treatment is required for the non-linear process of
atmospheric chemistry, since the δ-concentrations are strongly influenced by the background
concentrations in such processes. Although this treatment involves taking the difference of two
concentration fields, it does not magnify the spurious oscillations (the Gibbs phenomenon), which
are primarily generated in the advection step. The non-linear effects can be accounted for in the δ-
concentrations without losing track of the contributions arising from the specific emission source or
sector.
2.1.2. Model evaluation
Surface concentrations modelled by the DEHM model were evaluated against data at selected urban
background and regional or global monitoring stations in each Nordic country. The statistical
comparisons included using correlation coefficient ($r$), mean bias ($MB$) and normalized mean bias
($NMB$) and root mean square error ($RMSE$). The station information is provided in Table S1, along
with the descriptions of the monitoring network in each country.
2.2. Economic Valuation of Air Pollution (EVA) System
The EVA system (Brandt et al., 2013a,b; Geels et al., 2015; Im et al., 2018a) is based on the
impact-pathway chain method (Friedrich and Bickel, 2001). The EVA system can estimate acute
(short-term) and chronic (long-term) mortality, related to acute exposure to $O_3$, and $SO_2$, and
chronic exposure to $PM_{2.5}$, and the associated external costs. The EVA system requires gridded
concentrations along with gridded population data, exposure-response functions (ERFs) for health
impacts, which are recommended by the WHO (2013), and economic valuation functions of the
impacts from air pollution. In addition, EVA uses population densities over fixed age intervals,
corresponding to babies (under one year), children (under 15), adults (above 15 and above 30), and
elderlies (above 65). The impacts of short-term exposure to $O_3$, and $SO_2$, and the long-term
exposure to $PM_{2.5}$ are well established. EVA uses the annual mean concentrations of $SO_2$, and
$PM_{2.5}$, while for $O_3$, it uses the SOMO35 metric that is defined as the annual sum of the daily
maximum of 8-hour running average over 35 ppb, following WHO (2013) and EEA (2017).
The health impacts are calculated using an ERF of the following form:
$R = \alpha \times \delta_c \times P$
where $R$ is the response of the mortality rate or the years of life lost (in cases or days), $\delta_c$ denotes
the pollutant concentration, $P$ denotes the affected share of the population, and $\alpha$ an empirically
determined constant for the particular health outcome. EVA uses ERFs that are modelled as a linear
function, which is a reasonable approximation for the region of interest in the present study, as
showed in several studies (e.g. Pope et al., 2000; the joint World Health Organization/UNECE Task
Force on Health (EU, 2004; Watkiss et al., 2005)). However, some studies showed non-linear
relationships, being steeper at lower than at higher concentrations (e.g. Samoli et al., 2005).
Therefore, linear relationships may lead to overestimated health impacts over highly polluted areas.
Exposure response functions (ERF) for all-cause chronic mortality due to $PM_{2.5}$ are based on Pope
et al., 2002; Krewski et al., 2009), which are also recommended by the WHO (2013). These are the
most extensive and up-to-date data, although there are ongoing studies in Europe, and in particular
in the Nordic region to develop regional-specific ERFs (e.g. the Nordic WelfAir project:
https://projects.au.dk/nordicwelfair/). The current version of the EVA system used in the present
study does not include impacts due to exposure to $NO_2$. However, a new version is currently under
development under the NordicWelfAir project.
EVA calculates the number of lost life years for a Danish population cohort with normal age
distribution, when applying the ERF of Pope et al. (2002) for all-cause mortality (relative risk, RR=
1.062 (1.040-1.083) on 95% confidence interval). The latency period sums to 1138 year of life lost
(YOLL) per 100 000 individuals for an annual $PM_{2.5}$ increase of 10 μg m$^{-3}$ (Andersen et al, 2008).
The YOLL is then converted to number of cases by dividing by 10.6 following Watkiss et al.
(2005). The counterfactual $PM_{2.5}$ concentration is assumed to be 0 μg m$^{-3}$ following the EEA
methodology, meaning that the impacts have been estimated for the simulated total (anthropogenic
and natural) $PM_{2.5}$ mass. Applying a low counterfactual concentration can underestimate health
impacts at low concentrations if the relationship is linear or close to linear (Anenberg et al., 2016).
However, it is important to note that uncertainty in the health impact results may increase at low
concentrations due to sparse epidemiological data. Assuming linearity at very low concentrations

may distort the true health impacts of air pollution in relatively clean atmospheres (Anenberg et al., 2016).

Regarding short-term exposure to $O_3$, EVA uses the ERF recommended by the CAFE Programme (Hurley et al., 2005) and WHO (2013) that uses the daily maximum of 8-hour mean $O_3$ concentrations. There are also studies showing that $SO_2$ is associated with acute mortality, and EVA adopts the ERF identified in the APHENA study – Air Pollution and Health: A European Approach (Katsouyanni et al., 1997). Some recent studies also report the chronic effects from $O_3$ (e.g. Turner, 2016), however the current version of the EVA model does not include these effects. The ERFs used in EVA to calculate mortality are presented in Table 2.

Table 2. Exposure-response functions (ERF) used in EVA to calculate premature mortality.

| Health effects (compounds) | Exposure-response coefficient | Valuation, € |
|---|---|---|
| | ($\alpha$) | (EU27) |
| Acute mortality[2,3] ($SO_2$) | 7.85E-6 cases/$\mu$gm$^{-3}$ | 1,532,099 per case |
| Acute mortality[2,3] ($O_3$) | 3.27E-6*SOMO35 cases/$\mu$gm$^{-3}$ | |
| Chronic mortality[1,4], YOLL (PM) | 1.138E-3 YOLL/$\mu$gm$^{-3}$ (>30 years) | 57,510 per YOLL |
| Infant mortality[5], IM (PM) | 6.68E-6 cases/$\mu$gm$^{-3}$ (< 9 months) | 2,298,148 per case |

[1] Pope et al. (2002), [2] Anderson (1996), [3] Touloumi (1996), [4] Pope et al. (1995), [5] Woodruff et al. (1997).

For the valuation of the health impacts, a value of EUR 1.5 million was applied for preventing an acute death, following expert panel advice (EC, 2001), while for the valuation of a life year, a value of EUR 57 500 per year of life lost (YOLL) were applied (Alberini et al., 2006). More details can be found in Im et al. (2018a).

2.3. Scenarios (response and contribution)

We have applied a 30% reduction on land-based anthropogenic emissions from each of the continental Nordic countries, which include Denmark, Finland, Norway and Sweden. Each simulation perturbed a SNAP sector from an individual Nordic country, which are listed in Table 3. Industry is perturbed as the combination of SNAP 3,4,5 and 6, while agriculture (SNAP9) and waste management (SNAP 10) are perturbed as one combined sector.

DEHM model has been run on "tagged" mode, explained in section 2.1., so each simulation included a "perturbed" and "non-perturbed" concentration, which we used to calculate the response to the 30% reduction in the particular country and sector. These responses are then converted to population-weighted contributions using the gridded population densities and by assuming a linear extrapolation to 100%.

Table 3. Source sectors used in the perturbation scenarios.

| Source Sectors | SNAP Code |
|---|---|
| Combustion in energy and transformation industries | 1 |
| Non-industrial Combustion | 2 |
| Industry | 3,4,5,6 |
| Road transport | 7 |
| Other mobile sources and machinery | 8 |
| Waste and agriculture | 9,10 |

3. Results and Discussion

3.1. Evaluation

Surface ozone and $PM_{2.5}$ concentrations calculated by the DEHM model have been evaluated using surface observations from the urban background and regional background monitoring stations in the Nordic countries. The comparison of the mean of all observed concentrations in each country and the corresponding modelled concentrations are presented in Table 4 while Figs. 2 and 3 present Taylor diagrams for each station in each Nordic country, giving insight to the spatial distribution of model performance. As seen in Table 3, temporal variation of $O_3$ levels are well reproduced by the DEHM model over all countries ($r > 0.6$), however with an overestimation of ~10% over Denmark, Finland and Sweden, and ~30% over Norway. The daily variations of $PM_{2.5}$ levels, averaged over all stations in each Nordic country are well reproduced for Denmark ($r>$~0.7), moderately over Norway and Sweden ($r>0.4$), and poorly ($r$~0) over Finland (Table 3). $PM_{2.5}$ concentrations are underestimated by up to 35% over Denmark, Finland and Norway, and overestimated by 8% over Sweden.

In all countries, lower *NMB* values are calculated for $O_3$ over the regional background stations compared to urban background stations, where values are overestimated. Regarding $PM_{2.5}$, no such conclusions can be drawn due to very limited number of regional background stations in Denmark and Norway. In Finland, lower *NMB* values for $PM_{2.5}$ are calculated for the regional background stations, while in Sweden, much lower *NMB* values are calculated for the urban stations. These differences reflect the underestimations in emissions as well as the coarse model resolution, as well as missing sources, in particular for PM, such as wind-blown and resuspended dust in the DEHM model. It should also be mentioned that the modelled PM does not contain residual water. Table S2 shows the same comparisons for $NO_2$ and $SO_2$. The underestimations in the modelled $PM_{2.5}$ levels imply an underestimated exposure to $PM_{2.5}$ levels, given the dominance of $PM_{2.5}$ in premature mortality. Similarly, the overestimations in $O_3$ levels can be attributed to the underestimated NO-titration (Table S2).

Table 4. Model evaluation for the daily mean concentrations of O$_3$ and PM$_{2.5}$ for all the selected
stations in the Nordic countries.

| | O$_3$ | | | | | PM$_{2.5}$ | | | | |
| | *r* | *Obs. (µg m$^{-3}$)* | *NMB (%)* | *NME (%)* | *RMSE (µg m$^{-3}$)* | *r* | *Obs. (µg m$^{-3}$)* | *NMB (%)* | *NME (%)* | *RMSE (µg m$^{-3}$)* |
|---|---|---|---|---|---|---|---|---|---|---|
| Denmark | 0.91 | 59.59 | 0.10 | 0.11 | 7.65 | 0.85 | 10.77 | -0.31 | 0.31 | 3.78 |
| Finland | 0.85 | 55.20 | 0.10 | 0.15 | 9.24 | 0.02 | 5.05 | -0.16 | 0.24 | 1.56 |
| Norway | 0.73 | 54.65 | 0.27 | 0.29 | 14.78 | 0.66 | 6.85 | -0.36 | 0.36 | 2.76 |
| Sweden | 0.86 | 57.88 | 0.13 | 0.15 | 9.49 | 0.35 | 5.00 | 0.08 | 0.30 | 1.62 |


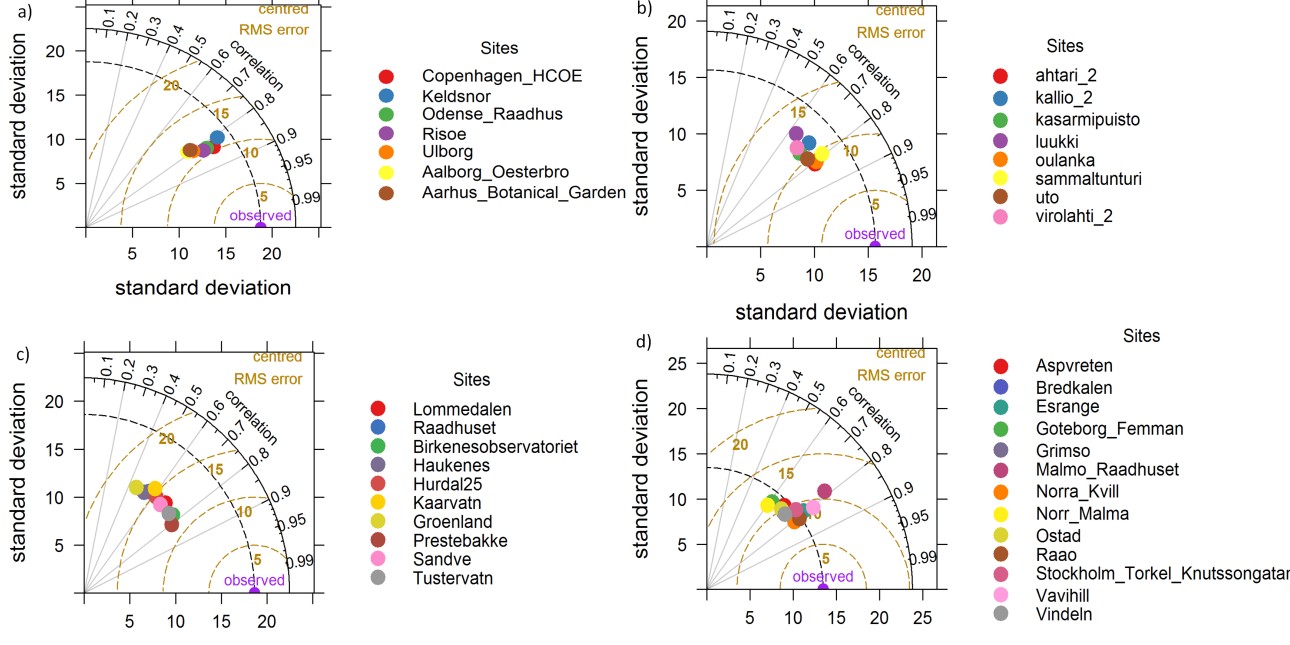

Figure 2. Taylor diagrams for daily mean O$_3$ for all stations in a) Denmark, b) Finland, c) Norway
and d) Sweden.

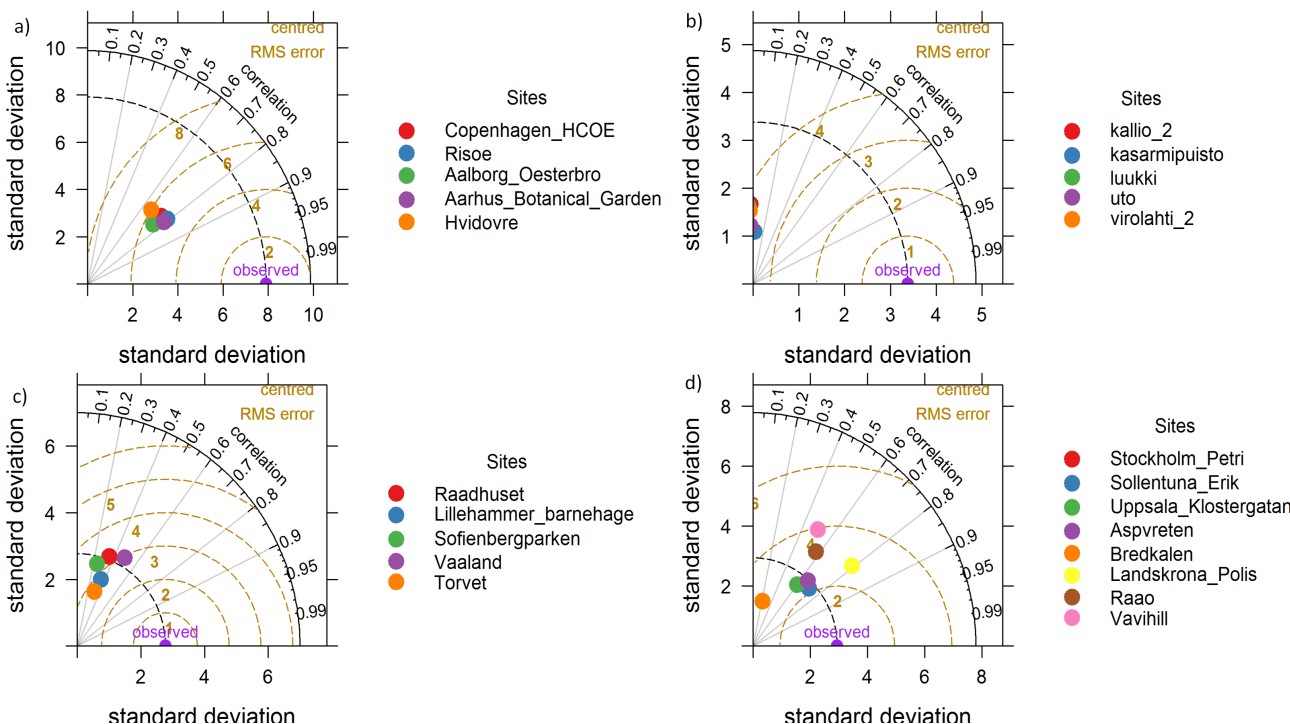

Figure 3. Taylor diagrams for daily mean PM$_{2.5}$ for all stations in a) Denmark, b) Finland, c) Norway and d) Sweden.

3.2. Sectoral contributions to surface concentrations

### 3.2.1. Nordic countries

In general, the long-term transport of air pollutants from one country to another is dependent on the global and regional atmospheric circulation and on the relative geographic positions of the countries. Nordic countries are influenced by substantial long-range transported contributions of air pollution especially from the central, western and central eastern parts of Europe. In the region containing the continental Nordic countries, the prevailing atmospheric flow directions near the ground surface are from the west, south-west and south. Based on the prevailing atmospheric circulation patterns, it is therefore to be expected that, e.g., the emissions in Denmark will have a relatively larger influence on the pollution levels in the other Nordic countries than those in Finland.

Our simulations show that PM$_{2.5}$ mass concentrations over the Nordic countries are dominated by nitrate aerosols (30% - 45 %) and sea-salt (30% - 50%). SO$_4$ aerosols contribute 10 to 15% of PM$_{2.5}$ concentrations while OC contributes by 8-11%, and BC by 2-4% of the PM$_{2.5}$ mass. As SO$_4$ and NO$_3$ aerosols include NH$_4$ in DEHM, results suggest that NH$_4$ aerosols contribute by more than half of the PM$_{2.5}$ mass over the Nordic countries. The annual mean surface PM2.5 concentrations for Denmark, Finland, Norway and Sweden are calculated to be 9.1 μgm$^{-3}$, 4.4 μgm$^{-3}$, 4.8 μgm$^{-3}$ and 5.8 μgm$^{-3}$, respectively. These values are in agreement with those reported by the EEA (2017), however underestimating by 12% (Denmark) up to 30% (Norway).

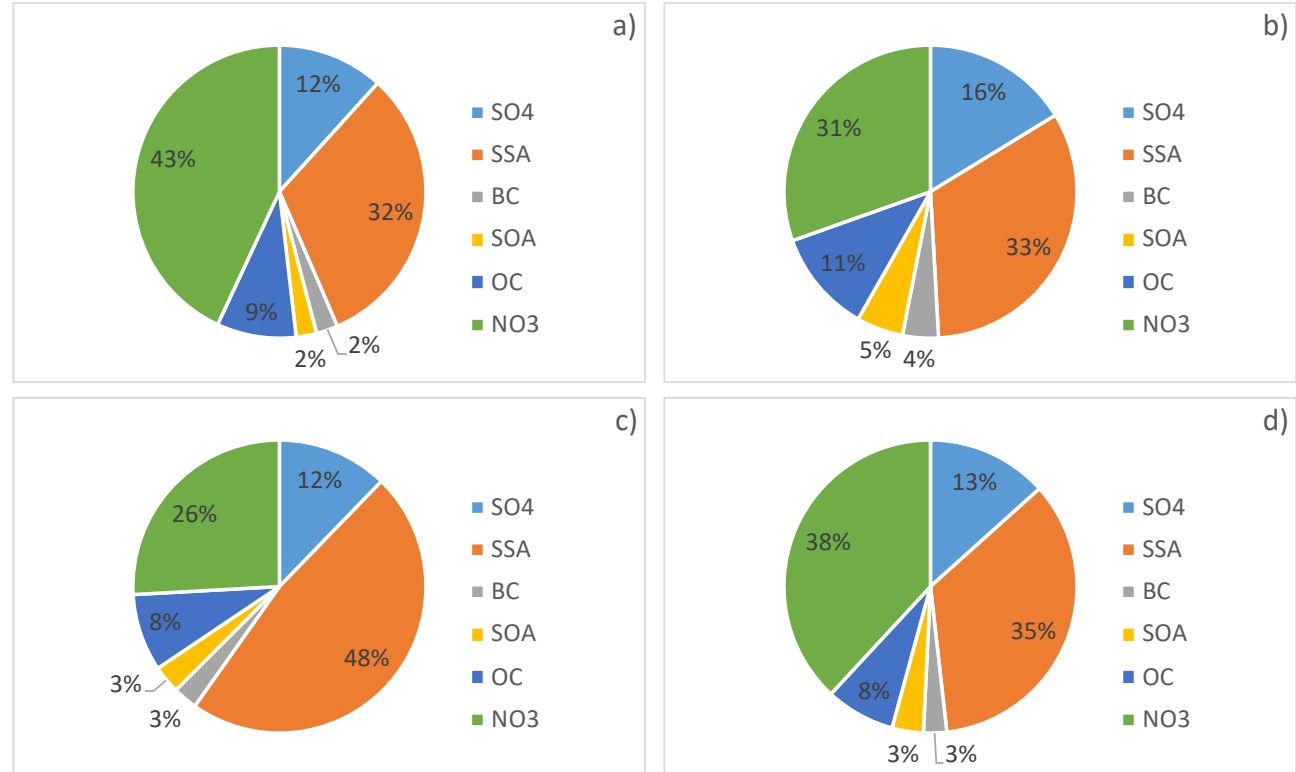

Figure 4. Simulated surface PM$_{2.5}$ chemical composition over a) Denmark, b) Finland, c) Norway, and d) Sweden.

Figure 5 compares the contribution of the total contributions anthropogenic sectors of each Nordic country on the surface concentrations over the country itself, with contributions from the anthropogenic sources in rest of the Nordic countries and rest of the world. Therefore, PM$_{2.5}$ in the figure does not contain the natural components that cannot be regulated, such as sea-salt. Figure 5 clearly shows that over 80% or more of PM$_{2.5}$ surface levels are transported outside the Nordic region, pointing that the Nordic countries are responsible for less than 20% of the particulate pollution in the region. This suggests significant decreases in the PM2.5 levels in the region can only be possible by reductions in the emissions upwind. Similar high contributions for other species including CO also shows that Nordic countries are exposed to airmasses coming from rest of the world while local pollution is low. The figure also shows that PM$_{2.5}$ levels are generally low in the Nordic countries, with annual means lower than 10 μg m$^{-3}$ (highest in Denmark and lowest in Finland). Similar to PM$_{2.5}$, annual mean surface O$_3$ levels are also low (~30 μg m$^{-3}$). Similar analyses done for O$_3$ (not shown) show that O$_3$ levels are controlled largely regional, where the local sources in the Nordic countries lead to small sink of O$_3$ due to NO-titration. This is also in agreement with Im et al. (2018b) reporting high Response to Extra-Regional Emission Reductions (RERER) values (>0.8) suggesting that O$_3$ is a regional background pollutant in Europe.


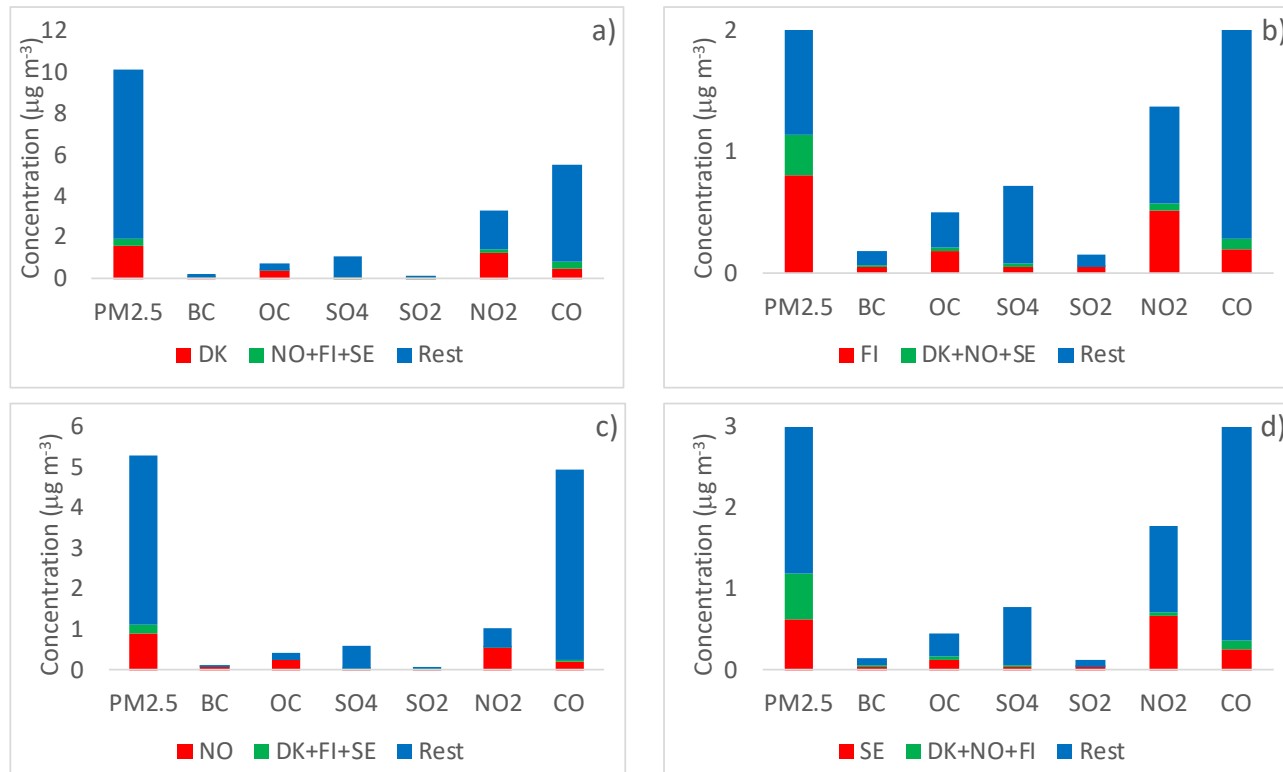

Figure 5. Absolute contributions of national, Scandinavian and other sources on the surface levels
of major air pollutants over a) Denmark, b) Finland, c) Norway and d) Sweden. Note that CO
concentrations are divided by 20 to scale with other pollutants.

Danish emissions contribute to only 1.14 µg m$^{-3}$ (13%) of the surface PM$_{2.5}$ concentrations over
Denmark (9.1 µg m$^{-3}$), while contributions to other Nordic countries are about 3% (Figure 6). Non-
industrial combustion (SNAP2), which is dominated by non-industrial combustion, is responsible
for 0.36 µg m$^{-3}$ (60%) of the Danish contribution to surface PM$_{2.5}$ concentrations over Denmark.
Non-industrial combustion contributes to 0.22 µg m$^{-3}$ (56%) of the Danish contribution to surface
organic carbon (OC) concentrations over the country, suggesting the importance of non-industrial
wood burning for heating. Industry contributes to 0.01 µg m$^{-3}$ (35%) of the Danish contribution to
the surface SO$_2$ concentrations over Denmark, while on-road and off-road transport contributes
equally to the Danish share of the in surface NO$_2$ concentrations by 1.02 µg m$^{-3}$ (~79% together).
Agriculture and waste handling are important sources for surface SO$_4$ levels over Denmark as well
as over the other Nordic countries, via the formation of ammonium sulfate ((NH$_4$)$_2$SO$_4$) due to the
large ammonia (NH$_3$) emissions from these sectors. 0.26 µg m$^{-3}$ of PM$_{2.5}$ over Denmark comes the
other Nordic countries, with 0.03 µg m$^{-3}$ coming from non-industrial combustion only.

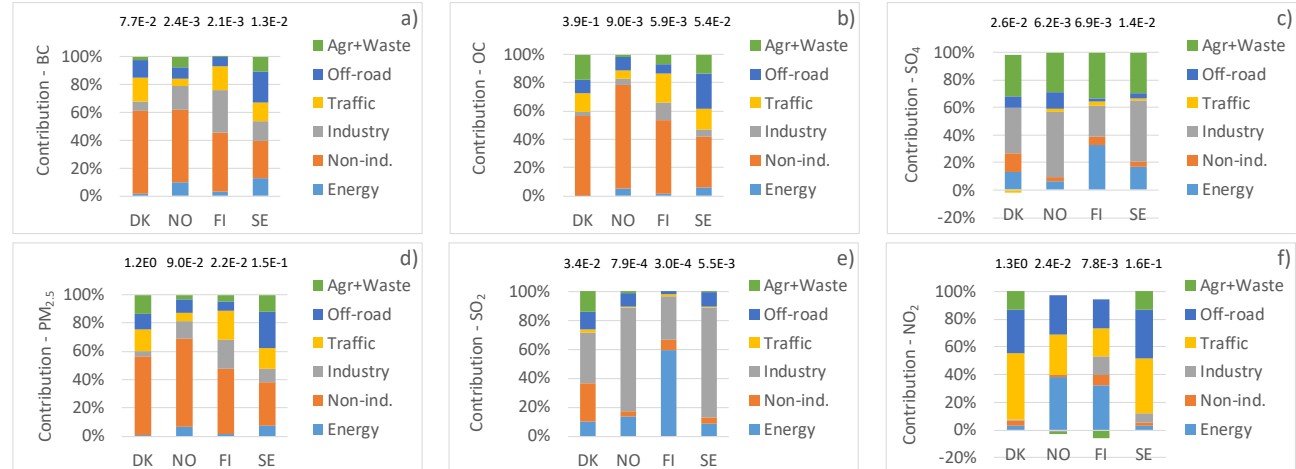

Figure 6. Population-weighted sectoral contributions of Danish emissions on surface a) BC, b) OC, c) SO$_4$, d) PM$_{2.5}$, e) SO$_2$ and f) NO$_2$ over the Nordic countries. The labels above the bars show the absolute total contribution in µg m$^{-3}$ from all the sectors in Denmark.

Contributions of the Norwegian emissions over the Nordic countries are presented in Figure 7. Similar to the Danish emissions, Norwegian emissions contribute to 0.6 µg m$^{-3}$ (13%) of the surface PM$_{2.5}$ concentrations over Norway, while contributions to other Nordic countries are below 1%, except for NO$_2$, where on-road transport emissions from Norway contributes to almost 0.02 µg m$^{-3}$ (42%) of the surface NO$_2$ levels over Finland. Non-industrial combustion is the main source of pollutant levels, in particular for OC, where Norwegian emissions are responsible for 0.18 µg m$^{-3}$ (74%) of local contribution to the surface OC levels over Norway. Industry is a major source of surface SO$_2$ levels over Norway, contributing to 0.02 µg m$^{-3}$ (66%) of the local contribution. 0.2 µg m$^{-3}$ of PM$_{2.5}$ levels over Norway comes from the other Nordic countries, 0.02 µg m$^{-3}$ being from non-residential combustion.

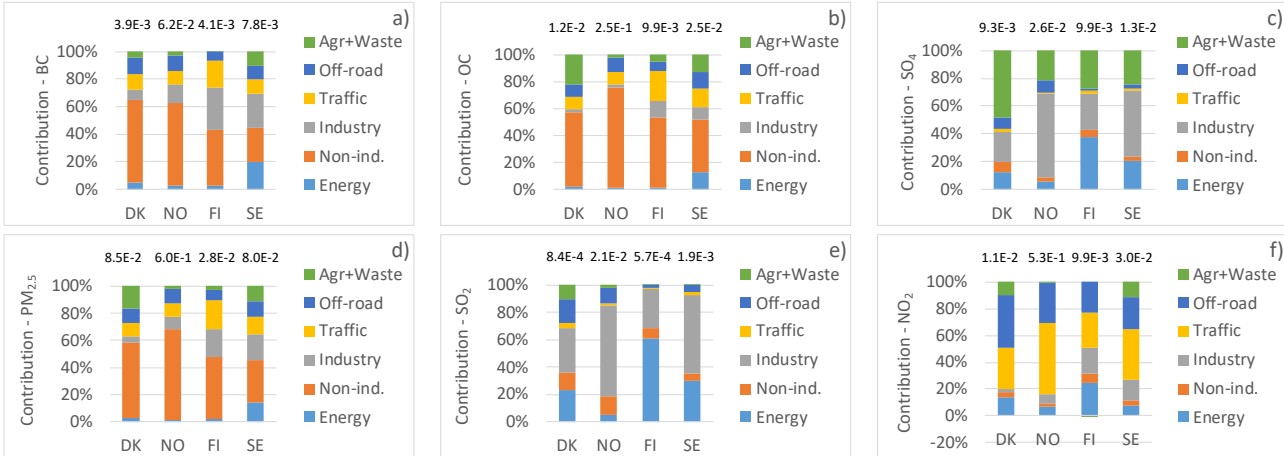

Figure 7. Population-weighted sectoral contributions of Norwegian emissions on surface a) BC, b) OC, c) SO$_4$, d) PM$_{2.5}$, e) SO$_2$ and f) NO$_2$ over the Nordic countries. The labels above the bars show the absolute total contribution in µg m$^{-3}$ from all the sectors in Norway.

Figure 8 shows the contributions of Finnish emissions on the pollutant levels over the Nordic countries. Similar to Denmark and Norway, non-industrial combustion is the major source of

pollution over Finland, although contributions are lower compared to Denmark and Norway (0.19 μg m$^{-3}$ (41%) of PM$_{2.5}$ and 0.11 μg m$^{-3}$ (48%) of OC). Another noticeable difference is that energy production is also an important contributor to surface SO$_2$ (0.01. μg m$^{-3}$: %44) and SO$_4$ (0.03 μg m$^{-3}$: 44%) levels over Finland. 0.3 μg m$^{-3}$ of PM$_{2.5}$ levels over Finland come from the other Nordic countries, 0.2 μg m$^{-3}$ being from non-residential combustion. Finnish emissions, in particular industrial combustion, contribute largest to the air pollution over Sweden.

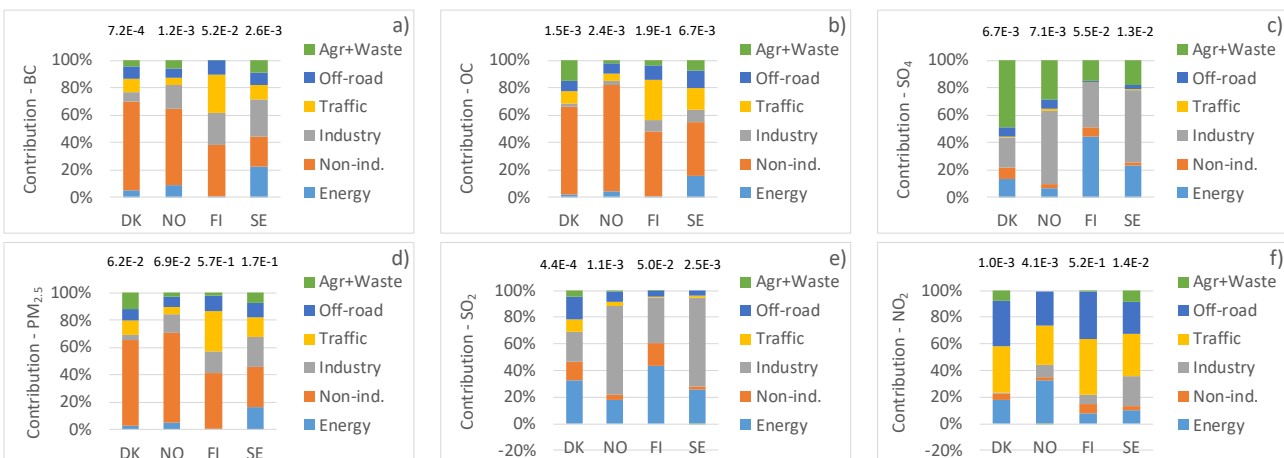

Figure 8. Population-weighted sectoral contributions of Finnish emissions on surface a) BC, b) OC, c) SO$_4$, d) PM$_{2.5}$, e) SO$_2$ and f) NO$_2$ over the Nordic countries. The labels above the bars show the absolute total contribution in μg m$^{-3}$ from all the sectors in Finland.

Contributions from the Swedish emission sources to surface pollutant levels over the Nordic countries are presented in Figure 9. Unlike other Nordic countries, Swedish emissions have larger contributions to pollution levels over the other Nordic countries, in particular over Norway. The figure also shows that Sweden does not experience as dominant contribution from non-industrial combustion (32%) like the other Nordic countries show. Swedish emissions from SNAP2 are much lower than for the rest of the Nordic countries (official emissions reported to the CLRTAP), most probably due to lower emission factors. Non-industrial combustion and industry contribute similarly to the surface PM$_{2.5}$ levels. Industry also has an important contribution to surface SO$_4$ levels (0.01 μg m$^{-3}$: 51%), as well to SO$_2$ (0.01 μg m$^{-3}$: 58%) and BC (0.006 μg m$^{-3}$: 18%). 0.5 μg m$^{-3}$ of surface PM$_{2.5}$ levels over Sweden comes from the other Nordic countries, of which, 0.1 μg m$^{-3}$ comes from non-residential combustion.

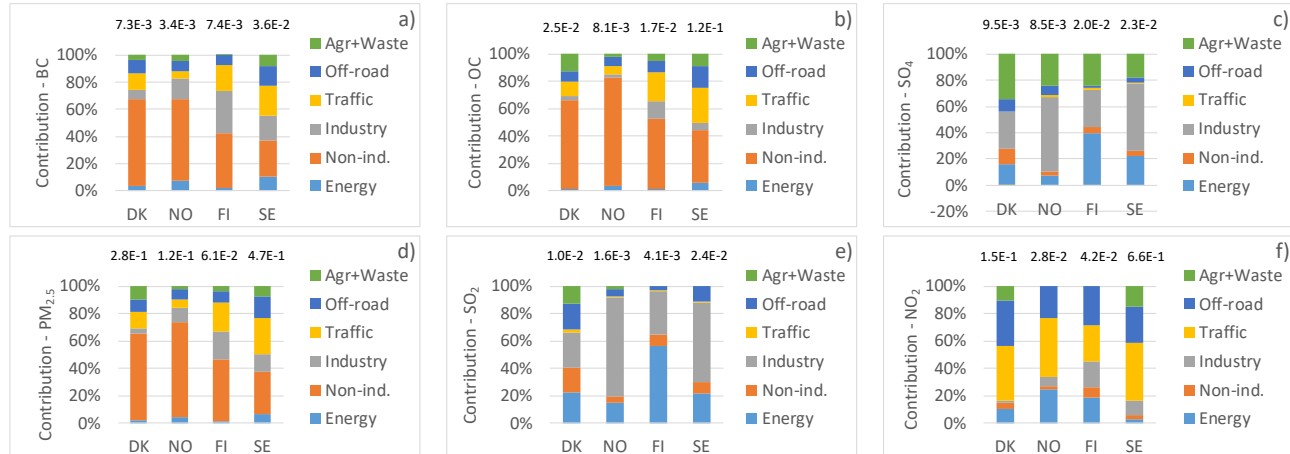

Figure 9. Population-weighted sectoral contributions of Swedish emissions on surface a) BC, b) SO$_4$, c) OC, d) PM$_{2.5}$, e) SO$_2$ and f) NO$_2$ over the Nordic countries. The labels above the bars show the absolute total contribution in µg m$^{-3}$ from all the sectors in Sweden.

### 3.2.2. Arctic

The contributions of the emission sources in the different Nordic countries on the surface aerosol concentrations over the Arctic region (defined as the area north of 67 °N latitude) are presented in Figure 10. Results show that overall, Norway has the largest contribution to surface aerosol levels over the Arctic, while Denmark has the lowest contribution, although contributions are only a few percent. Norwegian emissions, in particular non-industrial combustion, contributes to about 2% of the surface BC levels over the Arctic. Non-industrial combustion in the Nordic countries is also the largest contributor to Arctic BC levels, except for Sweden, where industry plays a more important role. Non-industrial combustion is also the dominant contributor to OC levels over the Arctic. Sulfate levels are largely influenced by the contributed from the agriculture and waste treatment facilities over the Nordic countries. Contributions to Arctic PM$_{2.5}$ levels are similar to the contributions to the BC levels.

476

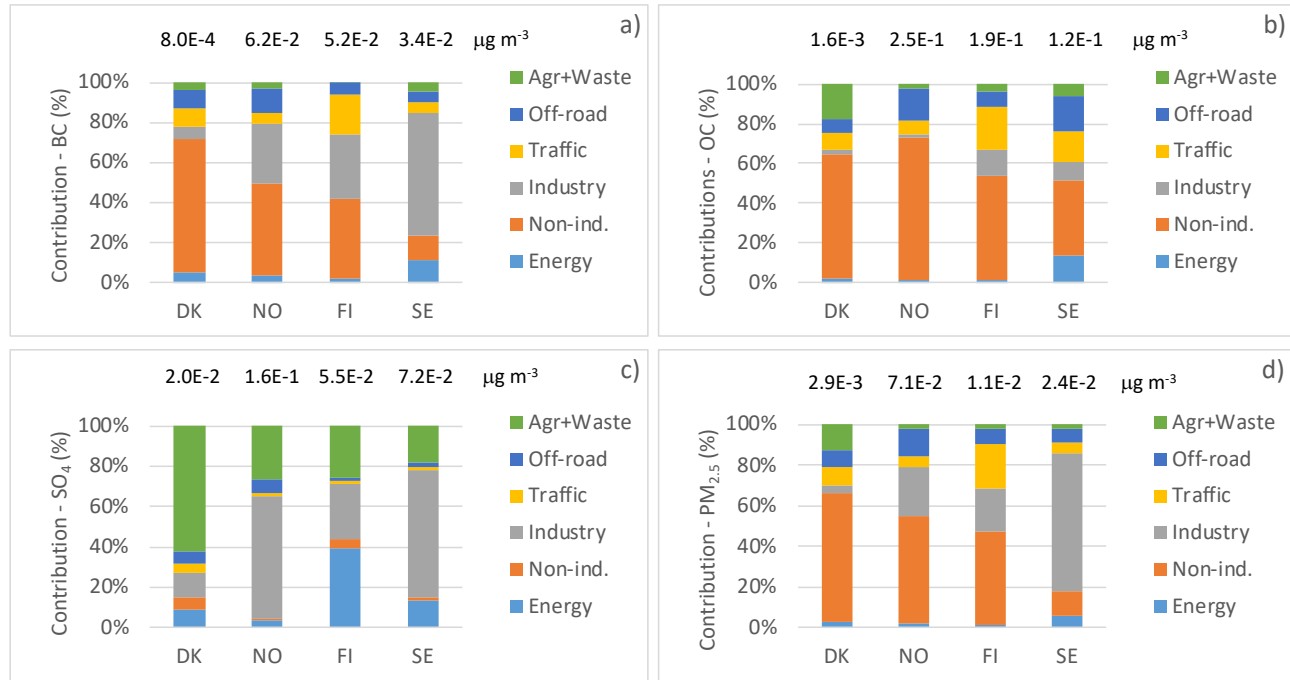

Figure 10. Population-weighted sectoral contributions from a) Denmark, b) Norway, c) Finland and d) Sweden to the surface aerosol levels over the Arctic (north of 67ºN). The labels above the bars show the absolute total contribution in μg m$^{-3}$ from all the sectors in each source country.

### 3.2.3. Spatial distributions of contributions

The geographical distributions of total anthropogenic emissions from each Nordic country to surface PM$_{2.5}$ and O$_3$ levels are calculated to investigate the extent of contributions from each Nordic country to its neighbours and to the Arctic. Figure 11 shows the annual-mean absolute contributions (%) of total land-based anthropogenic emissions to surface O$_3$ levels in the Nordic region from each country. The annual-mean contributions are very low, (up to 1.5 μg m$^{-3}$: 5%). Largest contributions in each country are calculated in the source region in the particular country, implying the impact of O$_3$ titration by local fresh NO emissions. Danish anthropogenic emissions (Figure 11a) lead to a titration of up to 1.5 μg m$^{-3}$ (around 4-5%), particularly over capital region. The largest impact of Finnish emissions is around the Helsinki area, responsible for up to 1 μg m$^{-3}$ (5%) of surface O$_3$ destruction over the area (Figure 11b). Finnish emissions also lead to an increase of surface O$_3$ levels by up to 0.5 μg m$^{-3}$ (1%) over the downwind regions to the southeast and northwest. Impact of Norwegian emissions to surface O$_3$ levels (Figure 11c) are largest (up to 1μg m$^{-3}$ : 2%) over the Oslo area and the impact extents over the northern part of Oslo with a slightly larger spatial contribution to O$_3$ levels compared to Denmark and Finland. The Swedish emissions have a larger geographical impact on the surface O$_3$ levels (Figure 11d) over the country itself compared to the other Nordic countries but the magnitude is similar to the impact from the Norwegian emissions.

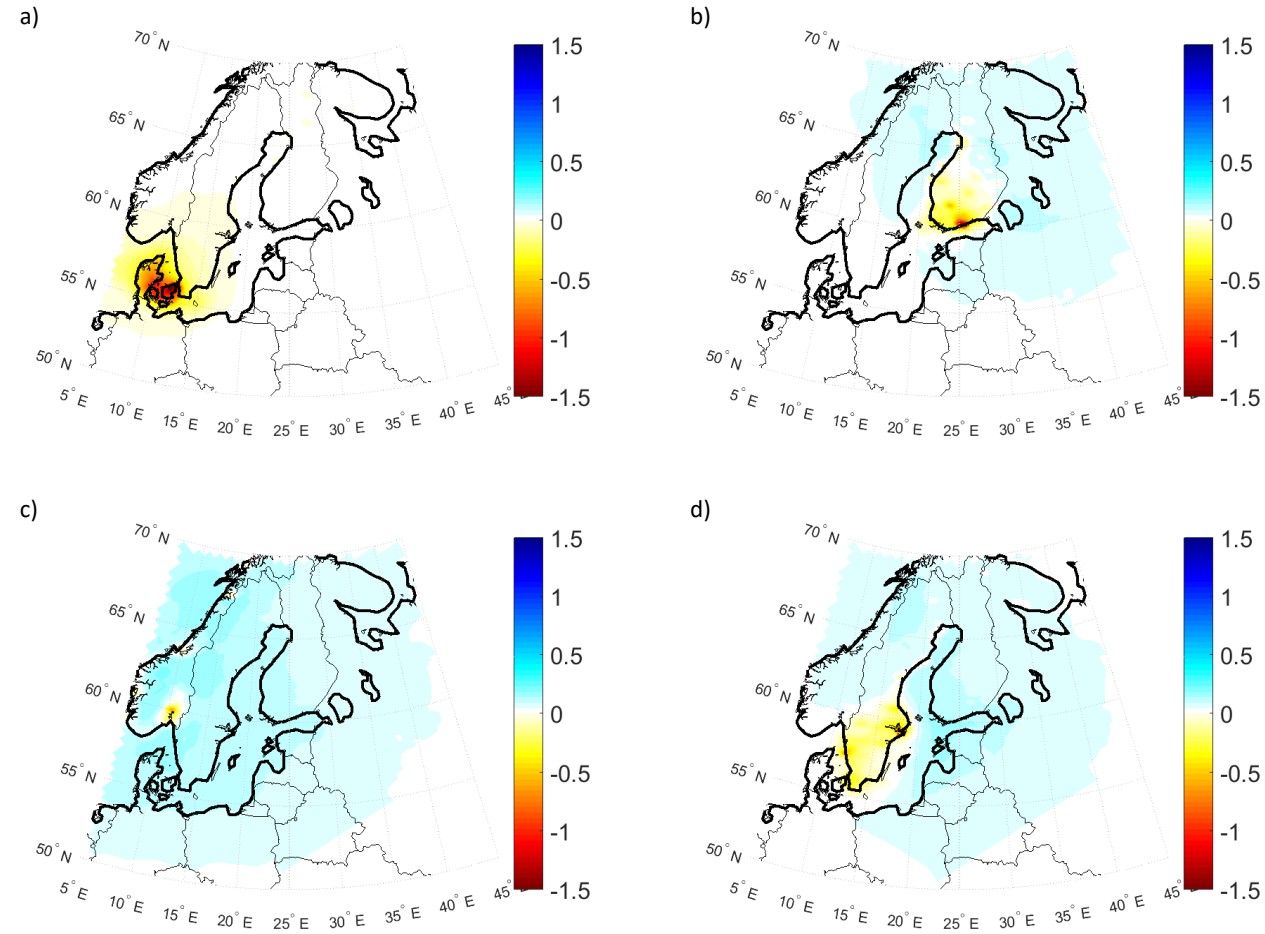

Figure 11. Spatial distributions of annual population-weighted mean absolute contributions ($\mu g\ m^{-3}$) of total emissions from a) Denmark, b) Finland, c) Norway, and d) Sweden to surface $O_3$ levels in the Nordic region.

Figure 12 shows the annual-mean absolute contributions of each Nordic country on the surface $PM_{2.5}$ levels in the entire model domain. Danish anthropogenic emissions are responsible for up to 20% of surface $PM_{2.5}$ levels over Denmark, with largest contributions over the capital region (Greater Copenhagen area) (Figure 12a). Danish land emissions also impact the surface $PM_{2.5}$ levels over the southern part of Sweden and Norway, by around 4% and 2%, respectively. The Finnish anthropogenic emissions have the largest impact on surface $PM_{2.5}$ levels over the southern part of the country, around the capital region by up to 30% (Figure 12b). Finnish emissions also have a small impact, lower than 3%, on the central part of Sweden and northern parts of Norway. Norwegian anthropogenic emissions have largest contributions to surface $PM_{2.5}$ level around the capital region by up to 30%, while there is also a significant impact on surface $PM_{2.5}$ levels over Sweden by around 7% (Figure 12c). Finally, Swedish anthropogenic emissions have large contribution to surface $PM_{2.5}$ levels over the Stockholm area by around 15% and also contributes to $PM_{2.5}$ levels over Finland, in particular over the southwestern parts of Finland, by up to 5% (Figure 12d).

Figure 12 also shows the impact of anthropogenic emissions from each Nordic country to the surface $PM_{2.5}$ over the Arctic. Overall, the impacts are very small, around a few per cent, as seen in

the figure. The Danish emissions (Figure 12a) have a more local contribution compared to other
Nordic countries and the impact does not reach above roughly 70 °N. The outflow from Finland,
Norway and Sweden can reach to the central Arctic ocean over to the northern parts of Greenland,
however contributions are around 1-2% (Figs. 12b-d).

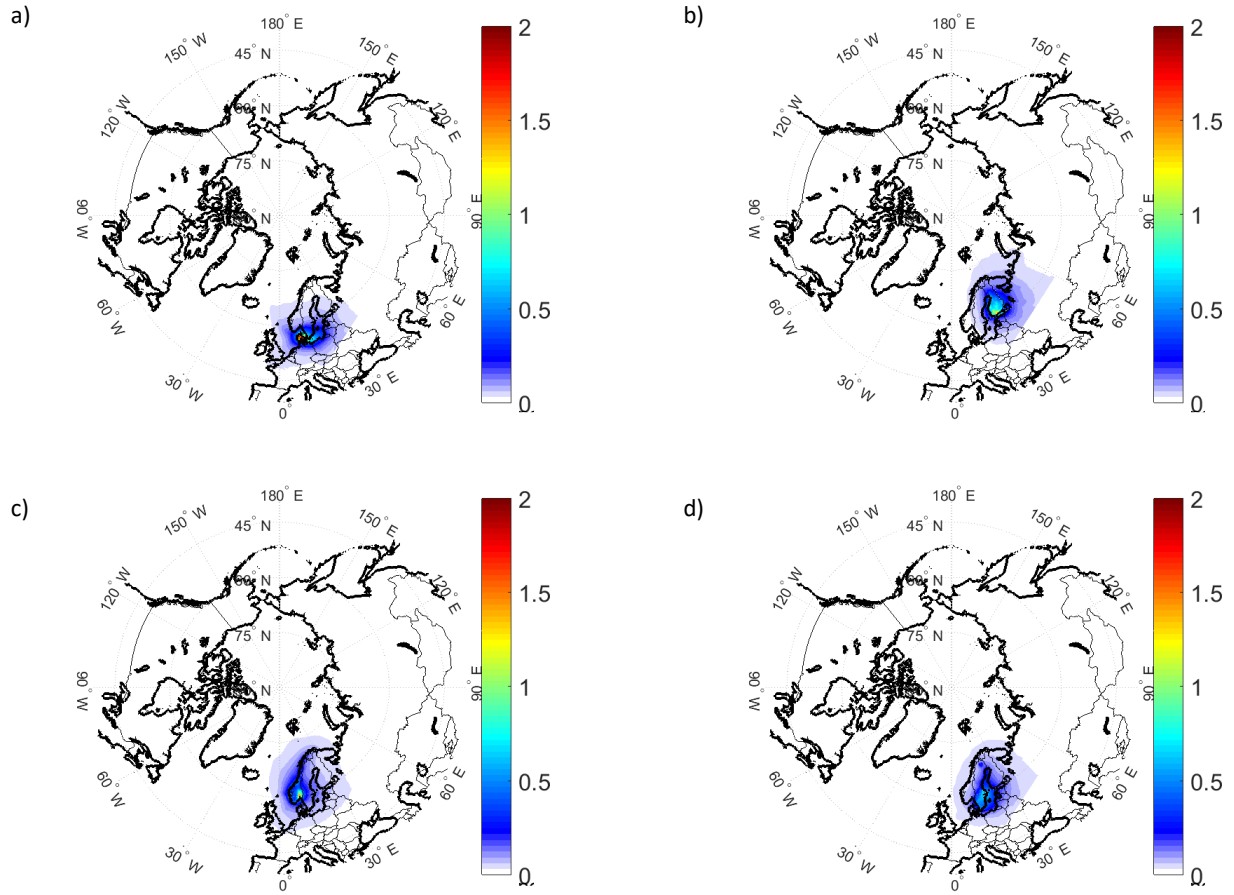

Figure 12. Spatial distributions of annual population-weigthed mean absolute contributions ($\mu g\ m^{-3}$)
of total emissions from a) Denmark, b) Finland, c) Norway, and d) Sweden to surface $PM_{2.5}$ levels
over the Nordic and the Arctic regions (north of 67ºN).
3.3.    Contribution to premature mortality and costs
The number of acute and chronic premature mortality in the four selected Nordic countries and the
Arctic region (north of 67ºN), along with the associated costs are presented in Table 5. The 95%
confidence intervals provided in the brackets are calculated by scaling the calculated health
outcomes by the confidence intervals of relative risk (RR) presented in section 2.2 (RR= 1.062
[1.040-1.083]). As seen in the Table, chronic mortality due to $PM_{2.5}$ is the major source for
premature mortality, as EVA calculates chronic mortality only due to exposure to $PM_{2.5}$ (see Table
2). The highest number of cases is calculated for Sweden (~4 200 cases), followed by Denmark
(~3 500 cases), Finland (~1 800) and Norway (~1 700). Results also show that $SO_2$ is responsible
for almost all acute mortalities in the region, which is consistent with earlier studies (e.g. Brandt et
al., 2013). This is due to the decrease of $O_3$ in the region by fresh NO emissions, leading to low
mortality due to $O_3$-exposure. These numbers lead to an associated cost of more than 2 billion Euros
in Sweden and Denmark and ~ 1 billion Euros in Finland and Norway. The number of premature
death cases are comparable with existing literature (e.g. Brandt et al., 2013a for Denmark; Solazzo
et al., 2018 for all four Nordic countries; EEA, 2017 for all four Nordic countries). In the Arctic
region, the total number of premature mortality cases is calculated to be 94, 93 of which are due to
exposure to $PM_{2.5}$ (chronic), leading to a cost of 58 million Euros.
Table 5. Acute and chronic premature death cases in the Nordic countries and the Arctic region
(north of 67ºN) in 2015 and the associated costs. The brackets show the 95% confidence interval.

| | Denmark | Finland | Norway | Sweden | Arctic |
|---|---|---|---|---|---|
| Premature Mortality (number of cases) | | | | | |
| Acute | 19 [19 20] | 18 [18 18] | 6 [6 6] | 25 [24 25] | 1 [1 1] |
| Chronic | 3 332 [3 263 3 398] | 1 707 [1 671 1 740] | 1 596 [1 563 1 628] | 4 091 [4 006 4 172] | 93 [91 95] |
| Total | 3 351 [3 282 3 417] | 1 725 [1 689 1 759] | 1 602 [1 569 1 634] | 4 115 [4 030 4 197] | 94 [92 96] |
| Cost (million Euros) | | | | | |
| Acute | 30 [29 30] | 28 [27 28] | 9 [9 10] | 38 [37 38] | 1 [1 1] |
| Chronic | 2 031 [1 989 2 071] | 1 040 [1 019 1 061] | 973 [953 992] | 2 494 [2 442 2 543] | 57 [56 58] |
| Total | 2 061 [2 018 2 102] | 1 068 [1 046 1 089] | 982 [962 1 002] | 2 531 [2 479 2 582] | 58 [57 59] |

The EVA model has been used to calculate the contributions of Nordic emissions to the total
premature mortality (acute + chronic) in the Nordic countries for the year 2015. Table 6 presents a
source/receptor matrix of the contributions to premature mortality on the Nordic countries. Danish
emissions contribute to ~400 premature deaths in Denmark, dominated by agriculture (33%), non-
industrial combustion (31%) and traffic (18%). In Norway, the dominating sector contributing is
non-industrial combustion, responsible for 48% of the ~200 premature deaths in Norway. In
Finland, the total number of premature deaths in 2015 is calculated to be ~270, where non-industrial
combustion and traffic are responsible for more than half. Finally, in Sweden, traffic and waste
management/agriculture are responsible for 50% of the total premature death in Sweden (~330).
Table 6. Source/Receptor relationships of the contributions of anthropogenic emissions from the
Nordic countries to the premature mortality in the Nordic area. The brackets show the 95%
confidence interval.

| Source/Receptor | Denmark | Finland | Norway | Sweden |
|---|---|---|---|---|
| Denmark | **422 [414 431]** | 24 [23 24] | 29 [28 29] | 198 [194 202] |
| Finland | 8 [8 8] | **274 [269 280]** | 9 [9 9] | 42 [41 43] |
| Norway | 33 [33 34] | 26 [26 27] | **203 [199 207]** | 86 [84 87] |
| Sweden | 57 [55 58] | 64 [63 65] | 27 [26 28] | **340 [333 346]** |

Figure 13 shows the contributions of sectoral emissions from each Nordic country to the total
premature death cases in 2015 in the different Nordic countries. Overall, Nordic countries
contribute to low premature death cases in their Nordic neighbours (≤50). As seen in the figure,
agriculture and waste management sectors together can have significant share in the premature
mortality (e.g. Denmark) due to the dominant contribution of $NH_4$ aerosols in the region (Figure 4).

The largest transboundary contribution is calculated for the Danish emissions, dominated by agriculture, non-industrial combustion and traffic, contributing to ~200 premature death cases in Sweden.

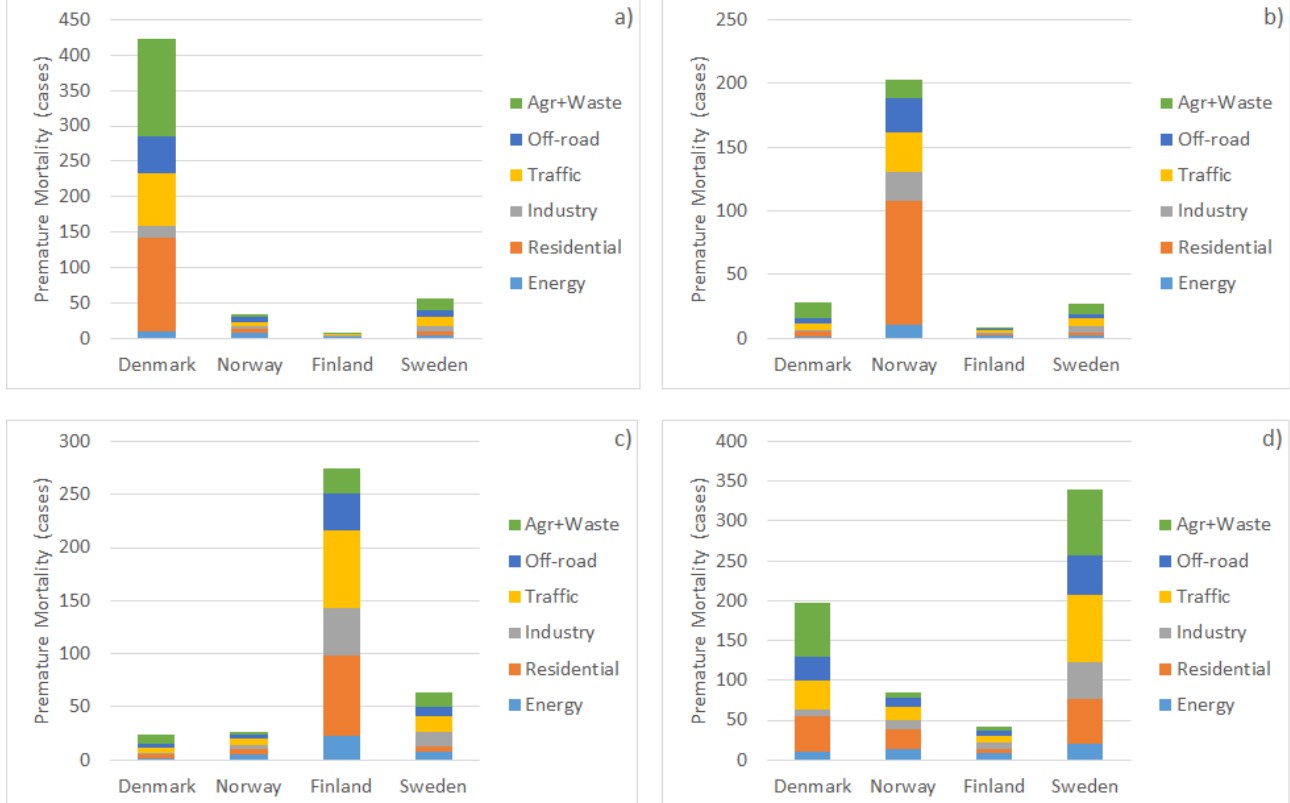

Figure 13. Source contributions from the anthropogenic emissions of a) Denmark, b) Norway, c) Finland, and d) Sweden to total premature mortality (acute+chronic) in the Nordic countries.

Table 7 shows the cost of air pollution on human health in each of the Nordic countries in the source country and the neighbouring Nordic countries. Among the four Nordic countries, Denmark has the largest external costs due to air pollution, followed by Sweden, Finland and Norway, respectively. Following the mortality rates, Denmark, Finland and Norway have the largest cost contribution to Sweden, while Sweden contributes largest to Denmark.

Table 7. Contribution of costs (million €) of air pollution impacts on human health in the Nordic
countries. The brackets show the 95% confidence interval.

| Source | Receptors | | | |
|---|---|---|---|---|
| | Denmark | Finland | Norway | Sweden |
| Denmark | 261 [256 266] | 14 [14 15] | 17 [17 18] | 122 [119 124] |
| Finland | 5 [5 5] | 172 [169 176] | 6 [5 6] | 26 [26 27] |
| Norway | 20 [20 21] | 16 [16 16] | 126 [123 128] | 53 [51 54] |
| Sweden | 36 [35 36] | 39 [39 40] | 17 [16 17] | 212 [207 216] |

Regarding the costs attributed to each of the source sectors, Figure S1 summarizes the contributions
per country. For Denmark, results suggest that non-industrial combustion and agriculture/waste
management are the main sectors to be targeted to reduce the negative impacts of air pollution. In
Norway, reduction of non-industrial combustion emissions alone can substantially reduce the costs
of air pollution. In Finland, similar to Denmark and Norway, non-industrial combustion should be
targeted for developing emission reduction strategies, along with the traffic emissions, which
contribute as large as the non-industrial combustion. Finally, in Sweden, traffic and
agriculture/waste management sectors should be targeted to reduce the adverse impacts of air
pollution and their associated costs. However, as the local contributions to air pollutants are
generally low in the region, it should be noted that significant reductions can only be achieved by
reducing the emissions upwind, which would require a coordinated effort in Europe.
4.    Conclusions
The sectoral contributions of land-based anthropogenic emission sources in the four Nordic
countries; Denmark. Finland, Norway and Sweden, on air pollution levels and premature mortality
in these countries and over the Arctic have been estimated using the DEHM/EVA impact
assessment system for the year 2015. The chemistry and transport model, DEHM, was run with
tagging mode in order to calculate inline the sectoral contributions based on 30% reductions of each
sector separately. Using the modelled surface concentrations of $O_3$, $SO_2$ and $PM_{2.5}$, the EVA model
calculated the acute ($O_3$ and $SO_2$) and chronic ($PM_{2.5}$) premature mortality due to exposure to these
pollutants.
Results show that the Nordic countries are responsible for 5-10% of the regional background
surface $PM_{2.5}$ concentrations in the countries itself. The non-industrial combustion (SNAP2), which
is dominated by the non-industrial wood combustion, is responsible for 50% to 80% of the
contribution to surface $PM_{2.5}$ in the Nordic countries. In Denmark, Finland and Norway, non-
industrial combustion contributes largely to surface OC (by 60% - 80%). In Sweden, SNAP2 is
responsible for 43% of the contribution to surface OC, while 43% comes from industrial activities.
Similar to OC, BC is also dominated by non-industrial combustion (by 50%-65%), except for
Sweden, where 25% originates from non-industrial combustion and 31% from industrial activities.
The dominant source for surface $SO_4$ and $SO_2$ in all four Nordic countries is calculated to be
industrial activities. In Norway and Sweden, around 70% of $SO_2$ are coming from industrial
activities, while in Denmark and Finland, industrial activities are responsible for around 30% of
$SO_2$. Off-road traffic is responsible for 21% of $SO_2$, while energy production is responsible for 50%
of $SO_2$ in Finland. Industrial activities are also responsible for 60% of $SO_4$ in Norway and Sweden
and 30% in Denmark and Finland. The dominant source for $NO_2$ is calculated to mobile sources,

and the share between on-road and off-road traffic varies depending on the country. Almost 35% of $NO_2$ comes from on-road traffic in all four Nordic countries while off-road traffic contributes by 25% to 35%.

Norway has the largest contribution to aerosol levels over the Arctic, while Denmark has the lowest contribution, although contributions are only a few percent. Non-industrial combustion in the Nordic countries is also the largest contributor to Arctic OC and BC levels, except for Sweden, where industry plays a more important role in relation to the Arctic levels. Agriculture and waste treatment facilities over the Nordic countries are responsible contribute to the sulfate levels over the Arctic.

Anthropogenic emissions lead to a titration of around 4-5%, particularly over the source countries and lead to a very small surface $O_3$ increase (>1%) in the downwind regions. The largest impacts are calculated to be around the capital regions. Danish emissions also impact the surface $PM_{2.5}$ levels over the southern part of Sweden and Norway, by around 3%. Finnish emissions also have a small impact, lower than 3%, on the central part of Sweden and northern parts of Norway. Norwegian anthropogenic emissions impacts $PM_{2.5}$ levels over Sweden by around 7% while Swedish anthropogenic emissions contribute to $PM_{2.5}$ levels over the southwestern parts of Finland, by up to 5%. It should be noted that these results are calculated for a specific year, 2015, therefore transport from one country to others can significantly vary in different years due to meteorology, in particular wind speed and direction.

The total number of premature mortality cases due to air pollution are calculated to be ~4 000 in Denmark and Sweden and ~2 000 in Finland and Norway, leading to a total cost of 7 billion Euros in the selected Nordic countries. The contributions of emission sectors to premature mortality in each of the Nordic countries vary. Danish agriculture and industrial emissions contribute similarly (by 33%) to ~400 premature mortality cases in Denmark, that are due to the Danish emissions. In Norway, non-industrial combustion, dominated by non-industrial wood combustion, is responsible for 48% of the ~200 premature deaths in Norway due to the exposure to pollution from the Nordic sources. In Finland, non-industrial combustion and traffic are responsible for more than half of the ~270 premature deaths in 2015, caused by the sources within the region. Finally, in Sweden, traffic and waste management/agriculture are responsible for 50% of the total premature death in Sweden (~330), caused by the emissions in the Nordic region. In Denmark, Finland and Norway, non-industrial combustion is the main sectors to be targeted to reduce the negative impacts of air pollution, while in Sweden, traffic and agriculture/waste management sectors should be targeted to reduce the adverse impacts of air pollution and their associated costs. Overall, Nordic countries contribute to low premature death cases in their Nordic neighbours (≤50). Among the four Nordic countries, Denmark has the largest external costs due to air pollution, followed by Sweden, Finland and Norway, respectively. Following the mortality rates, Denmark, Finland and Norway have the largest cost contribution to Sweden, while Sweden contributes largest to Denmark.

Overall, results from the estimates of pollution export, premature mortality and associated costs suggest that in the Nordic countries, non-industrial combustion, which is dominated by non-industrial wood combustion, together with industry and traffic are the main sectors to be targeted for emission mitigation strategies. The contributions of emissions from Nordic countries to each other are small (≤10%), and to the Arctic (up to 2%), meaning that large reductions can be achieved only by coordinated efforts to decrease emissions in the upwind countries.

**Author Contribution**

UI and JHC conducted the model simulations. JHC and OKN worked with the emissions input. MS and RM contributed to the experimental design of the model simulations. UI, JK, CA and SL-A extracted measurement data from Denmark, Finland, Sweden and Norway, respectively. CG and JB contributed to premature mortality and cost calculations. All co-authors contributed to the manuscript.

**Acknowledgements**

This study has been conducted under the FREYA project, funded by the Nordic Council of Ministers, Climate and Air Pollution Group (grant agreement no. MST-227-00036). AU gratefully acknowledges the NordicWelfAir project funded by the NordForsk's Nordic Programme on Health and Welfare (grant agreement no. 75007). The work has also been funded by the Academy of Finland within the project GLOROIA and by the Research Council of Norway under the project BlackArc (contract no 240921).

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

Notes-468+STR, 2008.

Solazzo, E., Riccio, A., Van Dingenen, R., and Galmarini, S.: Evaluation and uncertainty estimation
of the impact of air quality modelling on crop yields and premature deaths using a multi-model
ensemble, Sci. Total Environ., 663, 1437–1452, 2018.

Touloumi, G., Samoli, E. and Katsuyanni, K. Daily mortality and "winter type" air pollution in
Athens, Greece - a time series analysis within the APHEA project. Journal of Epidemiology and
Community Health, 50 (suppl 1), S47 - S51, 1996.

Turner, M.: Long-Term Ozone Exposure and Mortality in a Large Prospective Study. Am. J.
Respir. Crit. Care Med. 193:1134–1142; doi: 10.1164/rccm.201508-1633OC, 2016.

Watkiss, P., Pye, S., and Holland, M.: Cafe CBA: Baseline Analysis 2000 to 2020. Service Contract
for Carrying out Cost-Benefit Analysis of Air Quality Related Issues, in Particular in the Clean Air
for Europe (Cafe) Programme, available at:
http://ec.europa.eu/environment/archives/cafe/activities/pdf/cba_baseline_results2000_2020.pdf
(last access: 29 February 2019), 2005.

WHO: 7 million premature deaths annually linked to air pollution, News release, World Health
Organization, available at: http:// www.who.int/mediacentre/news/releases/2014/air-pollution/en/
(last access: 26 February 2019), 2014.

WHO: Health risks of air pollution in Europe – HRAPIE: Recommendations of concentration-
response functions for cost-benefit analysis of particulate matter, ozone and nitrogen dioxide,
World Health Organization, available at:
http://www.euro.who.int/__data/assets/pdf_file/0006/238956/Health_risks_air_pollution_HRAPIE_
project.pdf?ua=1 (last access: 14 March 2019), 2013.

Woodruff, T.J., Grillo, J. and Schoendorf, K.C. The relationship between selected causes of
postneonatal infant mortality and particulate air pollution in the United States. Environmental
Health Perspectives, 105, 608-612, 1997.