# Peer review of "Contributions of Nordic anthropogenic emissions on air pollution and premature mortality over the Nordic region and the Arctic"

_Atmospheric Chemistry and Physics, 2019_

## Referee Comment (RC1) · Anonymous Referee #1 · 13 May 2019

This work described the contributions of Nordic anthropogenic emissions on air pollution and premature mortality over the Nordic region and the Arctic. Although this study provides important results and is well written, there remain some concerns in the current manuscript. First, one issue is that the results are unsatisfactory discussed with only a few references in the case of premature mortality. Are the results coherent to other studies, years, countries? Which are the limitations of your method? Major comments: L113: Why 2015? Justify the selected year. L231: Why you use exposure-response functions for chronic mortality based on data from United States from the year 2002? Didn't you find any study from your Nordic region? What is about short-term exposure for PM2.5? Are the results from United States really applicable in the

Nordic region? How could influence the use of ERF from different studies? Why you use short-term effects only for specific pollutants and for others only chronic mortality? I recommend including table S2 as regular table, as only with this table the paragraph is understandable. The paragraph is not very clear. I would also include some limitation here about the use of exposure-response functions or discuss the limitation. Explain and justify the reason for selecting the specific exposure-response functions from these studies. Fig. 5-9: I think you should use stacked percent plot, which could help to compare sectorial emissions. In order to see the total, you could label each bar. In the actual form, it is not possible to compare correctly some sectorial categories. L448-: Why only PM2.5? NOx? SO2? O3? Please indicate the confidence interval for the mortality estimations, particularly in Table 4. The same for the cost estimations.

Minor comments: L77: please use capital letter for Primary PM2.5 (PPM2.5) L79: "secondary inorganic PM2.5 (SIA)" should it be secondary inorganic aerosol? L123: DHEM, although the abbreviation was defined in the abstract here you should repeat it for the reader. The same for EVA in point 2.2. It is not advisable to use abbreviations as titles. Fig1: Please use a better graphical representation. For example, treemap, circular packing or something else in which the graph is easier to read. L206: It should be "European Environmental Agency (EEA)". L232: remove last ")". L305: "eacvh" -> each; "Figure 4" If you use figure reference in text at beginning, it would be better the style "Figure X". Please, revise the whole document. L307: "The figure" -> the bars or graph?; subindex missing for PM2.5

---

## Referee Comment (RC2) · Anonymous Referee #3 · 16 May 2019

In this modelling study, PM and O3 exposure and associated premature mortalities in the Nordic countries have been attributed to anthropogenic emission sources (sectors) in each of the countries. The attribution is based on the tagging methodology which is more accurate than linearized source-receptor relations, commonly used under conditions of limited CPU resources.

General comments:

The material obtained from the modelling is potentially relevant, however my feeling is that the results are not optimally evaluated and presented, and too much attention is given to less relevant issues. In my opinion, the major conclusion of this study is

that 80-85% percent of the air pollution impacts in the Nordic countries are coming from sources outside that region. This observation is reported in the abstract and in the conclusions, however without giving any further consideration on the (policy) relevance of this finding. It basically means that national air quality measures in the Nordic country apparently barely can contribute to improving air quality. Given this outcome, one can wonder what is the relevance of making a detailed discussion of individual other Nordic countries contributions by sector to PM2.5 in the receptor country (i.e. attribution by country and by sector of the tiny orange part in the bar graphs of Fig. 4). Based on Figure 4, the key question is: which source regions and sectors are then contributing to the gray portion of the stacked bars? Hence, if the aim of the study is "to identify emission sectors that should be targeted for mitigation to decrease air pollution levels in the Nordic countries" (L108) the authors have clearly overlooked the major contributing factors. If the authors still want to focus specifically on the Nordic countries' contribution only, this should be better motivated and framed in the introduction. (Besides, I also wonder what is the motivation for a separate health impact assessment for specifically the low-populated Arctic region >67°N).

Specific comments:

1. Throughout the text: use subscript in chemical names and PM2.5

2. L34 – 37 and throughout the paper: please use consistent naming for the sectors in text and figure legends; "non-industrial" and "residential" are interchanged. It is better to use just one consistent name (preferably "residential"). Same for "Agriculture and Waste" and "Others" (preferably use "Agriculture + Waste")

3. L33: '. . .80% of the PM2.5 concentration was attributed to transport from outside. . .'

4. L34-35: If 80% of PM2.5 comes from outside, how can residential combustion (inside the country) be responsible for 60% of total PM2.5 mass?

5. L38 OC is said to be the major contributor. In the main section however nothing

has been mentioned on the chemical mass balance of the PM2.5 in each country, and further ammonium nitrate has never been mentioned. Was this compound considered as a PM2.5 constituent? Earlier studies have suggested that ammonium nitrate is the dominant PM2.5 component in NW Europe (e.g. Lelieveld et al., 2015).

6. L39: if the tagging method was used, the contribution of the residential sector should not be 'suggested' from the chemical composition but result directly from the tagged species & sector?

7. L62: in Sweden originates

8. L66: instead of 'geographic' maybe better use 'spatial'?

9. It is unclear to me how a country contributes to x% of European PM2.5. Do you mean that the European-wide population-weighted PM2.5 concentration (i.e. exposure) contains x% PM2.5 emitted in that country (L77-79)? Please formulate more precisely.

10. L94: Can you be more quantitative on how 'low' air pollution levels are, e.g. relative to WHO target levels? Does 'low' refer to country-area-mean or population exposure? Does this also apply to urban locations?

11. L96-104: not sure if the climate impact of BC is relevant in this context. Define 'SLCFs' (L103)

12. L109 'decrease'

13. DEHM: What is the time resolution of the model output for the pollutants? Are O3 values produced at 1h time steps? Does the model include natural PM components (in particular seasalt)?

14. L151: Actually the large contribution of NH3 in 'Others' is due to Agriculture (only).

15. Figure 1 is difficult to read and would benefit from a different layout (e.g. stacked bar plots, and using a different Y-scale for each component). Alternatively, as the

total values are already given in the table, the plot could show the different relative contributions by sector (stacked bar or pie plots).

16. L165: The tagging method keeps track. . .

17. L167: What do you mean by "background concentrations"? Usually this refers to concentrations in absence of anthropogenic emissions. L168: not clear what is meant by "in parallel". In general this paragraph is rather difficult to understand for those not really confident with the tagging method. Would it be possible to include some mathematical equation(s) that express the basic principle(s)?

18. Model evaluation: L181: these data are not shown in the supplemental material (I presume they are presented in Tale 3 instead). Is the model resolution of 50kmx50km high enough to reproduce urban background concentrations? Does the model include a sub-grid treatment to simulate the urban increment (for PM) or titration decrement (for O3)? Table S1 should include as well station data for PM2.5, SOMO35, SO2 (i.e. all the metrics used for the health impact assessment). The references to data sources for the 4 countries can easily be moved to Table S1 so the section from L184 – L214 can be shortened.

19. Section 2.2 (EVA) could contain some more information on the exposure-response functions and RRs used both for PM and O3. Table S2 should be explained better. ERFs are commonly expressing the relative risk. It is strange to see the exposure-response coefficient expressed as mortalities per concentration unit. There must be an underlying calculation, involving exposed population number and baseline mortalities. Please expand this. Why is SO2 included as a risk but not NO2? WHO (2013, HRAPIE project) recommends PM, O3 and NO2 as major risk factors, but not SO2 and CO. It is not clear if CO was considered here: CO ERF is not mentioned in Table S2, but section 2.2 mentions it is included in EVA.

20. L225: "EVA calculates and uses annual mean. . ." change to: "EVA uses annual mean. . ."

21. Are natural PM components included in the PM2.5 exposure? L235: "full range of anthropogenic concentrations"?

22. L247-248: not clear if the perturbations were done for each individual SNAP sector, or for the combined sectors in case of Industry and Others

23. L253-255: what is the outcome of this comparison between 100% and scaled 30% perturbation?

24. Table 3 should also include the mean values for O3 and PM2.5. What about the model evaluation for SO2 and NOx?

25. Use consistent symbols for the correlation coefficient (either r or R)

26. Do the Taylor diagrams add essential new information to what is given in Table 3? What can be concluded from the diagrams that is not emerging from the table? Is model performance better/worse for specific station type?

27. L270-274: the discussion does not correspond to what is given in Table 3: r is not >0.7 for Norway. If the % overestimation refers to the NME, then for O3 it is rather 20% for Denmark, Finland and Sweden, not 10%. For PM2.5, the R for Norway in the table is 0.35, not 0.7. A relatively good R does not necessarily imply a good model performance: mean biases of the order of -3$\mu$g/m$^3$ are significant as can be seen in the NME (underestimation of 40 to 50% for all countries). Could this be due to a natural component in the measurements that was not considered in the model? What can be concluded from this model evaluation in the context of the further analysis? How robust are the model results?

28. L301: "near de ground surface": change to "near the surface"

29. L301: 'Caused by the..' change to "Based on the prevailing . . ."

30. L305 eacvh: each

31. Why does Fig. 4 not include NH4 and NO3? The sum BC+OC+SO4 is much lower

than total PM2.5. What makes up the remaining PM2.5 mass? If NO2 and CO are not used in the impact evaluation why show the values?

32. Why is the same analysis not made for O3 or SOMO35?

33. L307-308: this is a very important observation and raises immediately the question about the sources of this major rest contribution. This observation should not be left undiscussed.

34. Please increase text font in Figures 5 to 7.

35. L324: 7 $\mu$g m-3

36. L328: again: what is the contribution of ammonium nitrate?

37. In Section 3.2.1, the text is quite repetitive, basically repeating for each combination of source country x and receptor country z the contribution for each chemical component. How relevant is this separation in components in the context of the formulated scope of this study (i.e. to identify the emission sectors that should be targeted for mitigation)? To answer this question it is more relevant to show for each of the receptor regions how much is being contributed (1) from the country's own emissions (2) from other Nordic countries (3) from the rest of the world (by difference). I would suggest to move Figs. 5-8 to the supplemental information. Instead, for each relevant exposure metric (PM2.5, O3 and maybe SO2, NO2, CO in the supplemental material), a figure could then be presented for each receptor country (DK, NO, FI, SE), with each bar representing a sector, and within each (stacked) bar a contribution from within the country, from other Nordic countries, and from the rest of the world (and maybe an additional bar for the sum of all sectors) as in attached figure 1 (made up with arbitrary numbers).

38. Similar comment for the Arctic (Fig. 9) where a graph could show the contributions by sector from each Nordic country and the rest of the world. But what is the relevance of considering specifically the >67°N area for health impacts? The contribution from

the Scandinavian countries are very low, also here it would be interesting to see what the major contributors to this receptor region are.

39. Are the concentrations and % shown in Figs 4 – 8 referring to exposure (i.e. population-weighted concentrations) or grid-area-weighted mean? To answer the formulated scientific question it should be exposure. For SE, NO and FI which have large portions of uninhabited area there could be a significant difference between area and population-weighted average.

40. If the graphs are produced as suggested, including PM2.5 and O3, the grid maps Fig 10 and 11 add little new information and they could be omitted (or transferred to the supplemental information)

41. If the grid maps are kept, please adapt the color scale of the O3 grid maps. Use the same range for the 4 maps, and make an upper limit that extends further above zero (now it seems that everything is colored red because the scale is cut off at a too low limit).

42. L405: what do you mean with "...are mainly calculated in the source country itself."

43. L 406 "Zealand region" has no meaning to a readership not familiar with the regional naming details.

44. L405 – 407 ("Danish anthropogenic. . .towards south") I can't follow the reasoning here: titration leads to a -4 to -5% contribution, but also to a +1% increase south? Also, as the scale stops at 0, this cannot be observed in graph 10a.

45. What is the share of O3 and SO2 in the acute mortalities?

46. L469: Given the fact that PM2.5 is the major risk factor in mortalities, why is the contribution of AGR so dominant in DK (compared to the small share in Fig 5)? Is this because the population exposure was taken into account? How is the share of sectors in the mortalities evaluated? By using the same proportion as in the population-weighted PM composition? In table 5 it would be useful to put in brackets which share

in total mortalities in each receptor country the numbers represent - e.g. 422 (13%).

47. L503 – 510: this is no new information because the costs are proportional to the mortalities for which it was already stated which sectors are dominating (L496 - 474). Further, when making recommendations on which sectors to address in order to "substantially reduce the costs of air pollution", the authors seem to have overlooked that 80 - 85% of the pollution health impact is imported from other regions.

48. L566 It is not 50% of total but 50% of premature deaths caused by the Nordic countries (the latter being 16% of total premature mortalities).

49. L579 -578: To my opinion this is the most relevant conclusion of this study. It leaves the reader with the feeling that the less relevant part of the data has been analyzed in too much detail, leaving this essential part untouched. . .

References

Lelieveld, J., Evans, J. S., Fnais, M., Giannadaki, D. and Pozzer, A.: The contribution of outdoor air pollution sources to premature mortality on a global scale, Nature, 525(7569), 367–371, doi:10.1038/nature15371, 2015.
* * *
[Figure]

[Figure]

Fig 1 Fictitious example for alternative format for Figs 5 - 8

**Fig. 1.**

---

## Author Comment (AC1) · 9 Aug 2019

We thank both reviewers for their valuable and constructive comments that helped us to improve the manuscript significantly. We list point by point responses to their comments below and hope they find the new version sufficient to be published in ACP.

Responses to comments from Reviewer#1

This work described the contributions of Nordic anthropogenic emissions on air pollution and premature mortality over the Nordic region and the Arctic. Although this study provides important results and is well written, there remain some concerns in the current manuscript. First, one issue is that the results are unsatisfactory discussed with only a few references in the case of premature mortality. Are the results coherent to other studies, years, countries? Which are the limitations of your method?

Major comments:

*Comment: L113: Why 2015? Justify the selected year.*

Response: We have now added the following (lines 115-118): "Year 2015 is selected to be in agreement with the ongoing Coupled Model Intercomparison Project Phase 6 (CMIP6: Eyring et al., 2016), where the current year is 2015. As the present study will also look at the impacts in the future using baseline scenarios from the CMIP6, we have selected the present year to be 2015 for consistency."

*Comment: L231: Why you use exposure-response functions for chronic mortality based on data from United States from the year 2002? Didn't you find any study from your Nordic region? What is about short-term exposure for PM2.5? Are the results from United States really applicable in the Nordic region? How could influence the use of ERF from different studies? Why you use short-term effects only for specific pollutants and for others only chronic mortality? I recommend including table S2 as regular table, as only with this table the paragraph is understandable. The paragraph is not very clear. I would also include some limitation here about the use of exposure-response functions or discuss the limitation. Explain and justify the reason for selecting the specific exposure-response functions from these studies.*

Response: We have now extended the health impact assessment section to discuss these concerns from the two reviewers.

*Comment: Fig. 5-9: I think you should use stacked percent plot, which could help to compare sectorial emissions. In order to see the total, you could label each bar. In the actual form, it is not possible to compare correctly some sectorial categories.*

Response: The figures have been modified accordingly.

*Comment: L448-: Why only PM2.5? NOx? SO2? O3? Please indicate the confidence interval for the mortality estimations, particularly in Table 4. The same for the cost estimations.*

Response: $NO_2$, $O_3$ and $SO_2$ are mostly associated with acute health impacts, although there are some recent studies suggesting chronic impacts from $O_3$. However, the current version of the EVA model does not yet include these impacts. This is now also added to the text (lines 226-231).

Minor comments:

*Comment: L77: please use capital letter for Primary PM2.5 (PPM2.5)*

Response: Corrected accordingly.

*Comment: L79: "secondary inorganic PM2.5 (SIA)" should it be secondary inorganic aerosol?*

Response: Corrected accordingly.

*Comment: L123: DHEM, although the abbreviation was defined in the abstract here you should repeat it for the reader. The same for EVA in point 2.2. It is not advisable to use abbreviations as titles.*

Response: Corrected accordingly.

*Comment: Fig1: Please use a better graphical representation. For example, treemap, circular packing or something else in which the graph is easier to read.*

Response: Fig.1. has been modified accordingly.

*Comment: L206: It should be "European Environmental Agency (EEA)".*

Response: Corrected accordingly.

*Comment: L232: remove last ")".*

Response: Removed.

*Comment: L305: "eacvh" -> each;*

Response: Corrected.

*Comment: "Figure 4" If you use figure reference in text at beginning, it would be better the style "Figure X". Please, revise the whole document.*

Response: the text is modified accordingly.

*Comment: L307: "The figure" -> the bars or graph?; subindex missing for PM2.5*

Response: We have now modified the text.

Responses to comments from Reviewer#2

In this modelling study, PM and O3 exposure and associated premature mortalities in the Nordic countries have been attributed to anthropogenic emission sources (sectors) in each of the countries. The attribution is based on the tagging methodology which is more accurate than linearized source-receptor relations, commonly used under conditions of limited CPU resources.

General comments:
*The material obtained from the modelling is potentially relevant, however my feeling is that the results are not optimally evaluated and presented, and too much attention is given to less relevant issues. In my opinion, the major conclusion of this study is that 80-85% percent of the air pollution impacts in the Nordic countries are coming from sources outside that region. This observation is reported in the abstract and in the conclusions, however without giving any further consideration on the (policy) relevance of this finding. It basically means that national air quality measures in the Nordic country apparently barely can contribute to improving air quality. Given this outcome, one can wonder what is the relevance of making a detailed discussion of individual other Nordic countries contributions by sector to PM2.5 in the receptor country (i.e. attribution by country and by sector of the tiny orange part in the bar graphs of Fig. 4).*

*Based on Figure 4, the key question is: which source regions and sectors are then contributing to the gray portion of the stacked bars? Hence, if the aim of the study is "to identify emission sectors that should be targeted for mitigation to decrease air pollution levels in the Nordic countries" (L108) the authors have clearly overlooked the major contributing factors. If the authors still want to focus specifically on the Nordic countries' contribution only, this should be better motivated and framed in the introduction. (Besides, I also wonder what is the motivation for a separate health impact assessment for specifically the low-populated Arctic region >67 N).*

Response: We thank the reviewer for the detailed comments. We have tried to address all the concerns in the following sections. The main aim of the study is to identify how much each Nordic country is contributing to the air pollution levels and exposure in the Scandinavian region and the Arctic. We agree that it is also very relevant and interesting how much is coming from rest of the world on a sectoral basis, however this is not the main objective of this study, and it requires additional sectoral simulations. It is also true that population-wise, the Arctic is not much relevant regarding health impact assessment, as it is for e.g. Europe. However, it is also interesting to have some first estimates of how much the Arctic population is affected from air pollution that is mainly transported from rest of the world. We tried to make this more clear in the end of the introduction section.

Specific comments:

*1. Throughout the text: use subscript in chemical names and PM2.5*

Response: Corrected accordingly.

*2. L34 – 37 and throughout the paper: please use consistent naming for the sectors in text and figure legends; "non-industrial" and "residential" are interchanged. It is better to use just one consistent name (preferably "residential"). Same for "Agriculture and Waste" and "Others" (preferably use "Agriculture + Waste")*

Response: We have modified. The text, the plots and the tables accordingly.

*3. L33: ': : :80% of the PM2.5 concentration was attributed to transport from outside: : :'*

Response: Corrected.

*4. L34-35: If 80% of PM2.5 comes from outside, how can residential combustion (inside the country) be responsible for 60% of total PM2.5 mass?*

Response: We mean that out of the 20% originating inside the region, 60 % is coming from non-industrial combustion. We have modified the sentence accordingly (lines 35-36).

*5. L38 OC is said to be the major contributor. In the main section however nothing has been mentioned on the chemical mass balance of the PM2.5 in each country, and further ammonium nitrate has never been mentioned. Was this compound considered as a PM2.5 constituent? Earlier studies have suggested that ammonium nitrate is the dominant PM2.5 component in NW Europe (e.g. Lelieveld et al., 2015).*

Response: We have now added a modelled PM2.5 chemical composition section under Section 3.2.1 (lines 345-352).

*6. L39: if the tagging method was used, the contribution of the residential sector should not be 'suggested' from the chemical composition but result directly from the tagged species & sector?*

Response: The tagging method identifies the contribution from all non-industrial combustion sources in the residential and commercial heating, including residential wood burning.

*7. L62: in Sweden originates*

Response: Corrected.

*8. L66: instead of 'geographic' maybe better use 'spatial'?*

Response: Changed accordingly.

*9. It is unclear to me how a country contributes to x% of European PM2.5. Do you mean that the European-wide population-weighted PM2.5 concentration (i.e. exposure) contains x% PM2.5 emitted in that country (L77-79)? Please formulate more precisely.*

Response: The cited study showed that Sweden contributed to 1.4% of the European Primary PM2.5 (PPM2.5) mass concentrations.

*10. L94: Can you be more quantitative on how 'low' air pollution levels are, e.g. relative to WHO target levels? Does 'low' refer to country-area-mean or population exposure? Does this also apply to urban locations?*

Response: We have modified the sentence accordingly (lines 95-97) as: "The Nordic countries are generally characterized among the EU countries with low air pollution levels (EEA, 2018). $PM_{2.5}$ levels are below the EU legislated limit value of 40 $\mu$g m$^{-3}$ as well as the WHO limit value of 20 $\mu$g m$^{-3}$ (EEA, 2018)."

*11. L96-104: not sure if the climate impact of BC is relevant in this context. Define 'SLCFs' (L103)*

Response: We have removed that sentence from the text.

*12. L109 'decrease'*

Response: Corrected accordingly.

*13. DEHM: What is the time resolution of the model output for the pollutants? Are O3 values produced at 1h time steps? Does the model include natural PM components (in particular seasalt)?*

Response: We have modified the section as (lines 138-141):

"The time resolution of the DEHM model is one hour. The gas-phase chemistry module includes 58 chemical species, 9 primary particles, including natural particles such as sea-salt and 122 chemical reactions (Brandt et al., 2012).",

and as (lines 143-148):

"In addition to the anthropogenic PM and SOA due to biogenic emissions, DEHM model also calculates sea-salt emissions and their transport and interactions with other pollutants. The current version of the DEHM model does not include wind-blown or re-suspended dust emissions. DEHM model does not output a PM2.5 or PM10 diagnostic, however these are calculated off-line, using all anthropogenic and natural components of PM, in order to be used in the health impact assessment described in Section 2.2."

*14. L151: Actually the large contribution of NH3 in 'Others' is due to Agriculture (only).*

Response: Modified accordingly.

*15. Figure 1 is difficult to read and would benefit from a different layout (e.g. stacked bar plots, and using a different Y-scale for each component). Alternatively, as the total values are already given in the table, the plot could show the different relative contributions by sector (stacked bar or pie plots).*

Response: We have changed Figure 1.

*16. L165: The tagging method keeps track: : :*

Response: Corrected accordingly.

*17. L167: What do you mean by "background concentrations"? Usually this refers to concentrations in absence of anthropogenic emissions. L168: not clear what is meant by "in parallel". In general this paragraph is rather difficult to understand for those not really confident with the tagging method. Would it be possible to include some mathematical equation(s) that express the basic principle(s)?*

Response: We have now modified the section. The technical details for the tagging scheme was already provided in Brandt et al. (2013), which is referred to in the manuscript..

*18. Model evaluation: L181: these data are not shown in the supplemental material (I presume they are presented in Tale 3 instead). Is the model resolution of 50kmx50km high enough to reproduce urban background concentrations? Does the model include a sub-grid treatment to simulate the urban increment (for PM) or titration decrement (for O3)? Table S1 should include as well station data for PM2.5, SOMO35, SO2 (i.e. all the metrics used for the health impact assessment). The references to data sources for the 4 countries can easily be moved to Table S1 so the section from L184 – L214 can be shortened.*

Response: 50 km resolution is coarse to reproduce the urban gradients. We have added some discussion for this in the text. This is why, we only do the comparison with urban and regional background stations. Data sources has been moved to the supplementary material.

*19. Section 2.2 (EVA) could contain some more information on the exposure-response functions and RRs used both for PM and O3. Table S2 should be explained better. ERFs are commonly expressing the relative risk. It is strange to see the exposure-response coefficient expressed as mortalities per concentration unit. There must be an underlying calculation, involving exposed population number and baseline mortalities.*

*Please expand this. Why is SO2 included as a risk but not NO2? WHO (2013, HRAPIE project) recommends PM, O3 and NO2 as major risk factors, but not SO2 and CO. It is not clear if CO was considered here: CO ERF is not mentioned in Table S2, but section 2.2 mentions it is included in EVA.*

Response: This section has been extended following the suggestions from both reviewers.

*20. L225: "EVA calculates and uses annual mean: : :" change to: "EVA uses annual mean: : :"*

Response: Modified accordingly.

*21. Are natural PM components included in the PM2.5 exposure? L235: "full range of anthropogenic concentrations"?*

Response: Changed accordingly (lines 238-239) as: "…simulated total (anthropogenic and natural) $PM_{2.5}$ mass.".

*22. L247-248: not clear if the perturbations were done for each individual SNAP sector, or for the combined sectors in case of Industry and Others*

Response: The perturbation is done for the combined snaps in case of Industry and Others.

*23. L253-255: what is the outcome of this comparison between 100% and scaled 30% perturbation?*

Response: We apologize for the error. This experiment was removed from the study and not conducted due to time limitation within the project.

*24. Table 3 should also include the mean values for O3 and PM2.5. What about the model evaluation for SO2 and NOx?*

Response: We have now added a Table S2 in the supplementary for the model evaluation for $NO_2$ and $SO_2$ and text in the manuscript (lines 308-313).

*25. Use consistent symbols for the correlation coefficient (either r or R)*

Response: Changed to *r* throughout the text.

*26. Do the Taylor diagrams add essential new information to what is given in Table 3? What can be concluded from the diagrams that is not emerging from the table? Is model performance better/worse for specific station type?*

Response: The Taylor diagrams show a station by station evaluation while the table gives an overall comparison based on the mean of stations in each country. In addition, we have added the following text (lines 310-316): "In all countries, lower NMB values are calculated for O3 over the regional background stations compared to urban background stations, where the overestimations are higher. Regarding PM2.5, no such conclusions can be drawn die to very limited number of regional background stations in Denmark and Norway. In Finland, lower NMB values for PM2.5 are calculated for the regional background stations, while in Sweden, much lower NMB values are calculated for the urban stations. These differences reflect the underestimations in emissions as well as the coarse model resolution."

*27. L270-274: the discussion does not correspond to what is given in Table 3: r is not >0.7 for Norway. If the % overestimation refers to the NME, then for O3 it is rather 20% for Denmark, Finland and Sweden, not 10%. For PM2.5, the R for Norway in the table is 0.35, not 0.7. A relatively good R does not necessarily imply a good model performance: mean biases of the order of -3g/m3 are significant as can be seen in*

*the NME (underestimation of 40 to 50% for all countries). Could this be due to a natural component in the measurements that was not considered in the model? What can be concluded from this model evaluation in the context of the further analysis? How robust are the model results?*

Response: We have now modified this section accordingly.

*28. L301: "near de ground surface": change to "near the surface"*

Response: Corrected

*29. L301: 'Caused by the.." change to "Based on the prevailing : : :"*

Response: Changed accordingly.

*30. L305 eacvh: each*

Response: Corrected.

*31. Why does Fig. 4 not include NH4 and NO3? The sum BC+OC+SO4 is much lower than total PM2.5. What makes up the remaining PM2.5 mass? If NO2 and CO are not used in the impact evaluation why show the values?*

Response: We have now added a paragraph for the modelled aerosol chemical composition (lines 353-360).

*32. Why is the same analysis not made for O3 or SOMO35?*

Response: We have done a similar analysis and the results are discussed in lines 376-379.

*33. L307-308: this is a very important observation and raises immediately the question about the sources of this major rest contribution. This observation should not be left undiscussed.*

Response: we have now added some discussion in the text (line 370-371).

*34. Please increase text font in Figures 5 to 7.*

Response: Modified accordingly.

*35. L324: 7 g m-3*

Response: Corrected.

*36. L328: again: what is the contribution of ammonium nitrate?*

Response: A section discussing chemical composition is now added to the text (lines 353-360).

*37. In Section 3.2.1, the text is quite repetitive, basically repeating for each combination of source country x and receptor country z the contribution for each chemical component. How relevant is this separation in components in the context of the formulated scope of this study (i.e. to identify the emission sectors that should be targeted for mitigation)? To answer this question it is more relevant to show for each of the receptor regions how much is being contributed (1) from the country's own emissions (2) from other Nordic countries (3) from the rest of the world (by difference). I would suggest to move Figs. 5-8 to the supplemental information. Instead, for each relevant exposure metric (PM2.5, O3 and maybe SO2, NO2, CO in the supplemental material), a figure could then be presented for each receptor country (DK, NO, FI, SE),*

*with each bar representing a sector, and within each (stacked) bar a contribution from within the country, from other Nordic countries, and from the rest of the world (and maybe an additional bar for the sum of all sectors) as in attached figure 1 (made up with arbitrary numbers).*

Response: We agree with the reviewer and indeed it is an interesting approach. However, this is not he main scope of the paper. The main focus is to calculate how much each Nordic country is responsible, on a sectoral basis, of their pollution in the Nordic and Arctic regions. In addition, this approach requires to make additional perturbation simulations for the present sectors for each country.

*38. Similar comment for the Arctic (Fig. 9) where a graph could show the contributions by sector from each Nordic country and the rest of the world. But what is the relevance of considering specifically the >67 N area for health impacts? The contribution from the Scandinavian countries are very low, also here it would be interesting to see what the major contributors to this receptor region are.*

Response: Please see above response.

*39. Are the concentrations and % shown in Figs 4 – 8 referring to exposure (i.e. population-weighted concentrations) or grid-area-weighted mean? To answer the formulated scientific question it should be exposure. For SE, NO and FI which have large portions of uninhabited area there could be a significant difference between area and population-weighted average.*

Response: This is a very good point and we thank the reviewer for raising this. All the numbers and plots are now updated to present the population-weighted contributions and concentrations.

*40. If the graphs are produced as suggested, including PM2.5 and O3, the grid maps Fig 10 and 11 add little new information and they could be omitted (or transferred to the supplemental information)*

Response. We keep the figures based on the responses to comments 37 and 38.

*41. If the grid maps are kept, please adapt the color scale of the O3 grid maps. Use the same range for the 4 maps, and make an upper limit that extends further above zero (now it seems that everything is colored red because the scale is cut off at a too low limit).*

Response: Maps are now modified.

*42. L405: what do you mean with "...are mainly calculated in the source country itself."*

Response: We have modified the sentence (lines 482-484) as: "The annual-mean contributions are very low, (up to 1.5 µg m-3: 5%). Largest contributions in each country are calculated in the source region in the particular country, implying the impact of O3 titration by local fresh NO emissions."

*43. L 406 "Zealand region" has no meaning to a readership not familiar with the regional naming details.*

Response: We have removed this.

*44. L405 – 407 ("Danish anthropogenic: : :towards south") I can't follow the reasoning here: titration leads to a -4 to -5% contribution, but also to a +1% increase south? Also, as the scale stops at 0, this cannot be observed in graph 10a.*

Response: We have now modified this sentence (lines 484-485) and the figure accordingly.

*45. What is the share of O3 and SO2 in the acute mortalities?*

Response: We have now added the following sentence (line 537-540): "Results also show that SO2 is almost responsible for all acute mortalities in the region, which is consistent with earlier studies (e.g. Brandt et al., 2013). This is due to the decrease of O3 in the region by fresh NO emissions, leading to low mortality due to O3-exposure."

*46. L469: Given the fact that PM2.5 is the major risk factor in mortalities, why is the contribution of AGR so dominant in DK (compared to the small share in Fig 5)? Is this because the population exposure was taken into account? How is the share of sectors in the mortalities evaluated? By using the same proportion as in the population-weighted PM composition? In table 5 it would be useful to put in brackets which share in total mortalities in each receptor country the numbers represent - e.g. 422 (13%).*

Response: We have now added the following sentence (line 568-570): "As seen in the figure, agriculture and waste management sectors can have significant share in the premature mortality (e.g. Denmark) due to the dominant contribution of NH4 aerosols in the region."

*47. L503 – 510: this is no new information because the costs are proportional to the mortalities for which it was already stated which sectors are dominating (L496 - 474). Further, when making recommendations on which sectors to address in order to "substantially reduce the costs of air pollution", the authors seem to have overlooked that 80 - 85% of the pollution health impact is imported from other regions.*

Response: We have moved the figure to the supplement. We have also added the following to the end of the paragraph (lines 600-602): "However, as the local contributions to air pollutants are generally low in the region, it should be noted that significant reductions can only be achieved by reducing the emissions downwind, which would require a coordinated effort in Europe."

*48. L566 It is not 50% of total but 50% of premature deaths caused by the Nordic countries (the latter being 16% of total premature mortalities).*

Response: we have modified as follows (lines 654-661): "Danish agriculture and industrial emissions contribute similarly (by 33%) to ~400 premature mortality cases in Denmark, that are due to the Danish emissions. In Norway, non-industrial combustion, dominated by non-industrial wood combustion, is responsible for 48% of the ~200 premature deaths in Norway due to the exposure to pollution from the Nordic sources. In Finland, non-industrial combustion and traffic are responsible for more than half of the ~270 premature deaths in 2015, caused by the sources within the region. Finally, in Sweden, traffic and waste management/agriculture are responsible for 50% of the total premature death in Sweden (~330), caused by the emissions in the Nordic region."

*49. L579 -578: To my opinion this is the most relevant conclusion of this study. It leaves the reader with the feeling that the less relevant part of the data has been analyzed in too much detail, leaving this essential part untouched: : :*

Response: Please refer to the response to the general comment and comment 37.

*References*
*Lelieveld, J., Evans, J. S., Fnais, M., Giannadaki, D. and Pozzer, A.: The contribution of outdoor air pollution sources to premature mortality on a global scale, Nature, 525(7569), 367–371, doi:10.1038/nature15371, 2015.*

---

## Referee Report (RR1)

Second review of

"Contributions of Nordic anthropogenic emissions on 1 air pollution and premature mortality over the Nordic region and the Arctic" by Im et al.

The authors have adequately addressed most issues raised in the first review, however there are still a number of inaccuracies and errors to be corrected.

Major issue:

Fig. 4 shows clearly the contribution of seasalt as a major PM2.5 component. Obviously, this contribution is not part of the considered anthropogenic sectors and cannot be regulated. It is not clear from the manuscript if the large 'Rest' contribution in Fig. 5 includes seasalt, or if only anthropogenic sectors are considered. This should be made clear in the text, and if seasalt is indeed included in Fig. 5, it sheds a different light on the 'external' contribution to PM2.5 and on the potential of Nordic versus external air quality control measures.

Minor comments:

L63: originating from south-western Europe

L. 96 – 97: The EU limit level for annual mean PM2.5 is 25µg/m³ (not 40). The WHO limit value is 10µg/m³ (not 20).

Section 2.1: Please mention whether residual water is included in the modelled PM2.5 mass or if it is dry PM2.5 (monitoring stations determine the mass at a standard RH of 50% which in theory would dry pure ammonium salts, however in reality the more complex chemical mixture retains some water).

Is there a difference between SO2 (table 1) and SOx (Figure 1)?

Section 2.2 (health impacts):

L217: c should be δc

L236: (Andersen, 2008) is missing in References

R in the equation (L216) should be specified as mortality rate or life years lost (per population); I don't see how it could represent 'days' or 'episodes'. The alpha values in Table 2 unit is cases/µgm-3/population (maybe it's more clear if the values are multiplied with a scaling factor 1E6 and expressed as cases/µgm-3/million population

I think SOMO35 should not be in the alpha formula in Table 2 (in the case of O3, SOMO35 = δc in the R equation).

Infant mortality: should be < 9 months instead of > 9 months in Table 2.

In this section, long-term health impact from PM is expressed in YOLL, however in Table 5 they are presented as number of premature deaths. Please make this consistent.

Section 3.1 (Evaluation): the authors did not address the request to include as well mean concentrations for O3 and PM2.5 in Table 4. Some values are given in the text in L357 – 359. This should be moved to the validation section and preferably introduced in the table. Same for model results for O3 (and NO2 and SO2 in the SI).

L311: where the overestimations are higher => where values are overestimated

L312: die to => due to

L 316-317 "Differences … can be attributed to … " : basically repeats what was written in L315.

Section 3.2:

L371: downwind => upwind (also in L602)

L485: leads => lead

Figure 11: please include country borders

L537: is almost responsible for => is responsible for almost

Table 5 and 6: What are the values in brackets? Describe in the caption. Presumably confidence intervals? Based on what? It's hard to believe that the model can predict mortalities with a 2% accuracy!

---

## Author Response (AR2)

Second review of "Contributions of Nordic anthropogenic emissions on 1 air pollution and premature mortality over the Nordic region and the Arctic" by Im et al.

The authors have adequately addressed most issues raised in the first review, however there are still a number of inaccuracies and errors to be corrected.

Response: We thank the reviewer for the comments. We have tried to answer all the issues raised by the reviewer now and hope that the manuscript is now suitable for publication.

Major issue:

*Comment: Fig. 4 shows clearly the contribution of seasalt as a major PM2.5 component. Obviously, this contribution is not part of the considered anthropogenic sectors and cannot be regulated. It is not clear from the manuscript if the large 'Rest' contribution in Fig. 5 includes seasalt, or if only anthropogenic sectors are considered. This should be made clear in the text, and if seasalt is indeed included in Fig. 5, it sheds a different light on the 'external' contribution to PM2.5 and on the potential of Nordic versus external air quality control measures.*

Response: Figure 5 does not include sea-salt. We have now updated the text accordingly (Lines 370-373).

Minor comments:

*Comment: L63: originating from south-western Europe*

Response: Modified.

*Comment: L. 96 − 97: The EU limit level for annual mean PM2.5 is 25µg/mÑ (not 40). The WHO limit value is 10µg/mÑ (not 20).*

Response: Corrected.

*Comment: Section 2.1: Please mention whether residual water is included in the modelled PM2.5 mass or if it is dry PM2.5 (monitoring stations determine the mass at a standard RH of 50% which in theory would dry pure ammonium salts, however in reality the more complex chemical mixture retains some water).*

Response: We have updated the text accordingly (Line 319).

*Comment: Is there a difference between SO2 (table 1) and SOx (Figure 1)?*

Response: We have corrected Table 1 accordingly.

Section 2.2 (health impacts):

*Comment: L217: c should be δc*

Response: Modified accordingly.

*Comment: L236: (Andersen, 2008) is missing in References*

Response: Reference is corrected and added to the list.

*Comment: R in the equation (L216) should be specified as mortality rate or life years lost (per population); I don't see how it could represent 'days' or 'episodes'. The alpha values in Table 2 unit is cases/μgm-3/population (maybe it's more clear if the values are multiplied with a scaling factor 1E6 and expressed as cases/μgm-3/million population. I think SOMO35 should not be in the alpha formula in Table 2 (in the case of O3, SOMO35 = δc in the R equation).*

Response: EVA model calculates also morbidity as well as mortality, including number of restricted days. However the present paper only considers mortality. We have modified the text accordingly (Line 217). We do keep the table as it is to be in agreement with previous publications.

*Comment: Infant mortality: should be < 9 months instead of > 9 months in Table 2.*

Response: This is corrected.

*Comment: In this section, long-term health impact from PM is expressed in YOLL, however in Table 5 they are presented as number of premature deaths. Please make this consistent.*

Response: We have now updated the text accordingly (Lines 237-238).

*Comment: Section 3.1 (Evaluation): the authors did not address the request to include as well mean concentrations for O3 and PM2.5 in Table 4. Some values are given in the text in L357 − 359. This should be moved to the validation section and preferably introduced in the table. Same for model results for O3 (and NO2 and SO2 in the SI).*

Response: We have now added the observed concentrations in the tables.

*Comment: L311: where the overestimations are higher => where values are overestimated*

Response: Modified accordingly.

*Comment: L312: die to => due to*

Response: Corrected.

*Comment: L 316-317 "Differences ... can be attributed to ... " : basically repeats what was written in L315.*

Response: We have now modified this part (Lines 316-319).

Section 3.2:

*Comment: L371: downwind => upwind (also in L602)*

Response: Corrected.

*Comment: L485: leads => lead*

Response: Corrected.

*Figure 11: please include country borders*

Response: Figures are updated.

*Comment: L537: is almost responsible for => is responsible for almost*

Response: Corrected.

*Comment: Table 5 and 6: What are the values in brackets? Describe in the caption. Presumably confidence intervals? Based on what? It's hard to believe that the model can predict mortalities with a 2% accuracy!*

Response: We have now updated the captions and the text accordingly (Lines 540-543).

[revised manuscript text omitted]

Font: 11 pt

| Page 9: [2] Formatted Table | Ulas Im | 19/09/2019 11:03:00 |
|---|---|---|

Formatted Table

| Page 9: [3] Formatted | Ulas Im | 19/09/2019 11:03:00 |
|---|---|---|

Font: 11 pt

| Page 9: [4] Formatted | Ulas Im | 19/09/2019 11:03:00 |
|---|---|---|

Font: 11 pt

| Page 9: [5] Formatted | Ulas Im | 19/09/2019 11:03:00 |
|---|---|---|

Font: 11 pt

| Page 9: [6] Formatted | Ulas Im | 19/09/2019 11:03:00 |
|---|---|---|

Font: 11 pt

| Page 9: [7] Formatted | Ulas Im | 19/09/2019 11:03:00 |
|---|---|---|

Font: 11 pt

| Page 9: [8] Formatted | Ulas Im | 19/09/2019 11:03:00 |
|---|---|---|

Font: 11 pt

| Page 9: [9] Formatted | Ulas Im | 19/09/2019 11:03:00 |
|---|---|---|

Font: 11 pt

| Page 9: [10] Formatted | Ulas Im | 19/09/2019 11:03:00 |
|---|---|---|

Font: 11 pt

| Page 9: [11] Formatted | Ulas Im | 19/09/2019 11:03:00 |
|---|---|---|

Font: 11 pt

| Page 9: [12] Formatted | Ulas Im | 19/09/2019 11:03:00 |
|---|---|---|

Font: 11 pt

| Page 9: [13] Formatted | Ulas Im | 19/09/2019 11:03:00 |
|---|---|---|

Font: 11 pt

| Page 9: [14] Formatted | Ulas Im | 19/09/2019 11:03:00 |
|---|---|---|

Font: 11 pt

| Page 9: [15] Formatted | Ulas Im | 19/09/2019 11:03:00 |
|---|---|---|

Font: 11 pt

| Page 9: [16] Formatted | Ulas Im | 19/09/2019 11:03:00 |
|---|---|---|

Font: 11 pt

| Page 9: [17] Formatted Table | Ulas Im | 19/09/2019 11:19:00 |
|---|---|---|

Formatted Table

| Page 9: [18] Formatted | Ulas Im | 19/09/2019 11:17:00 |
|---|---|---|

Font: (Default) Times New Roman, 11 pt

| Page 9: [19] Formatted | Ulas Im | 19/09/2019 11:18:00 |

Centred

| Page 9: [20] Formatted | Ulas Im | 19/09/2019 11:17:00 |

Font: 11 pt

| Page 9: [21] Formatted | Ulas Im | 19/09/2019 11:18:00 |

Font: (Default) Times New Roman, 11 pt

| Page 9: [22] Formatted | Ulas Im | 19/09/2019 11:19:00 |

Centred

| Page 9: [23] Formatted | Ulas Im | 19/09/2019 11:18:00 |

Font: 11 pt

| Page 9: [24] Formatted | Ulas Im | 19/09/2019 11:03:00 |

Font: 11 pt

| Page 9: [25] Formatted | Ulas Im | 19/09/2019 11:17:00 |

Font: (Default) Times New Roman, 11 pt

| Page 9: [26] Formatted | Ulas Im | 19/09/2019 11:18:00 |

Centred

| Page 9: [27] Formatted | Ulas Im | 19/09/2019 11:17:00 |

Font: 11 pt

| Page 9: [28] Formatted | Ulas Im | 19/09/2019 11:18:00 |

Font: (Default) Times New Roman, 11 pt

| Page 9: [29] Formatted | Ulas Im | 19/09/2019 11:19:00 |

Centred

| Page 9: [30] Formatted | Ulas Im | 19/09/2019 11:18:00 |

Font: 11 pt

| Page 9: [31] Formatted | Ulas Im | 19/09/2019 11:03:00 |

Font: 11 pt

| Page 9: [32] Formatted | Ulas Im | 19/09/2019 11:17:00 |

Font: (Default) Times New Roman, 11 pt

| Page 9: [33] Formatted | Ulas Im | 19/09/2019 11:18:00 |

Centred

| Page 9: [34] Formatted | Ulas Im | 19/09/2019 11:17:00 |

Font: 11 pt

| Page 9: [35] Formatted | Ulas Im | 19/09/2019 11:18:00 |

Font: (Default) Times New Roman, 11 pt

| Page 9: [36] Formatted | Ulas Im | 19/09/2019 11:19:00 |
|---|---|---|

Centred

| Page 9: [37] Formatted | Ulas Im | 19/09/2019 11:18:00 |
|---|---|---|

Font: 11 pt

| Page 9: [38] Formatted | Ulas Im | 19/09/2019 11:03:00 |
|---|---|---|

Font: 11 pt

| Page 9: [39] Formatted | Ulas Im | 19/09/2019 11:17:00 |
|---|---|---|

Font: (Default) Times New Roman, 11 pt

| Page 9: [40] Formatted | Ulas Im | 19/09/2019 11:18:00 |
|---|---|---|

Centred

| Page 9: [41] Formatted | Ulas Im | 19/09/2019 11:17:00 |
|---|---|---|

Font: 11 pt

| Page 9: [42] Formatted | Ulas Im | 19/09/2019 11:18:00 |
|---|---|---|

Font: (Default) Times New Roman, 11 pt

| Page 9: [43] Formatted | Ulas Im | 19/09/2019 11:19:00 |
|---|---|---|

Centred

| Page 9: [44] Formatted | Ulas Im | 19/09/2019 11:18:00 |
|---|---|---|

Font: 11 pt